# An atrial fibrillation-associated regulatory region modulates cardiac *Tbx5* levels and arrhythmia susceptibility

**Fernanda M Bosada**[1,2]**, Karel van Duijvenboden**[1]**, Alexandra E Giovou**[1]**, Mathilde R Rivaud**[1,2]**, Jae-Sun Uhm**[1,3]**, Arie O Verkerk**[1,2]**, Bastiaan J Boukens**[1,4]**, Vincent M Christoffels**[1]*****

[1]Department of Medical Biology, Amsterdam Cardiovascular Sciences, Amsterdam Reproduction and Development, Amsterdam University Medical Centers, University of Amsterdam, Amsterdam, Netherlands; [2]Department of Experimental Cardiology, Amsterdam Cardiovascular Sciences, Amsterdam University Medical Centers, University of Amsterdam, Amsterdam, Netherlands; [3]Department of Cardiology, Severance Hospital, College of Medicine, Yonsei University, Seoul, Republic of Korea; [4]Department of Physiology, University of Maastricht, Cardiovascular Research Institute Maastricht, Maastricht University Medical Center, Maastricht, Netherlands

**\*For correspondence:**
v.m.christoffels@amsterdamumc.nl

**Competing interest:** The authors declare that no competing interests exist.

**Abstract** Heart development and rhythm control are highly Tbx5 dosage-sensitive. *TBX5* haploinsufficiency causes congenital conduction disorders, whereas increased expression levels of *TBX5* in human heart samples has been associated with atrial fibrillation (AF). We deleted the conserved mouse orthologues of two independent AF-associated genomic regions in the *Tbx5* locus, one intronic (RE(int)) and one downstream (RE(down)) of *Tbx5*. In both lines, we observed a modest (30%) increase of *Tbx5* in the postnatal atria. To gain insight into the effects of slight dosage increase in vivo, we investigated the atrial transcriptional, epigenetic and electrophysiological properties of both lines. Increased atrial *Tbx5* expression was associated with induction of genes involved in development, ion transport and conduction, with increased susceptibility to atrial arrhythmias, and increased action potential duration of atrial cardiomyocytes. We identified an AF-associated variant in the human RE(int) that increases its transcriptional activity. Expression of the AF-associated transcription factor *Prrx1* was induced in *Tbx5*^RE(int)KO^ cardiomyocytes. We found that some of the transcriptional and functional changes in the atria caused by increased *Tbx5* expression were normalized when reducing cardiac *Prrx1* expression in *Tbx5*^RE(int)KO^ mice, indicating an interaction between these two AF genes. We conclude that modest increases in expression of dose-dependent transcription factors, caused by common regulatory variants, significantly impact on the cardiac gene regulatory network and disease susceptibility.

## Editor's evaluation

Molecular mechanisms of atrial fibrillation, a highly prevalent arrhythmia, have been challenging to contextualize. The investigators, in a series of experiments using the deletion of two loci in TBX5 (one intronic and one downstream), achieved a modest increase in Tbx5 in the atria. In a series of elegant experiments they show that this increase in TbX5 in turn impacted the cardiac gene regulatory network that resulted in a higher susceptibility to AF. These results will provide useful insights into the mechanisms of disease especially the variable phenotypes observed with functional variants of a gene.

## Introduction

The lifetime risk of developing a common disease, such as cardiovascular or neurodegenerative conditions, is influenced by genetic predisposition resulting from large numbers of inherited common genetic variants (single-nucleotide polymorphisms, SNPs). Disease-associated variants are typically found in noncoding genomic regions and are thought to affect the functionality of regulatory elements (REs) such as enhancers or elements involved in chromatin conformation (*Degtyareva et al., 2021*; *Deplancke et al., 2016*; *Hormozdiari et al., 2018*; *Maurano et al., 2012*; *Schaub et al., 2012*). These regulatory variants are pleiotropic and their effects on target gene expression are often specific to particular cell-types, conditions or stages of development (*GTEx Consortium, 2020*; *Sobreira et al., 2021*; *Strober et al., 2019*; *Watanabe et al., 2019*). It remains challenging to identify the causal variants among the many associated variants in a disease-associated noncoding DNA region and the REs that are affected by such variants (*Hormozdiari et al., 2017*). Moreover, common variants typically have a small effect on phenotype, and the different functional variants may act additively, synergistically, or oppositely. As a consequence, very few biological mechanisms linking disease-associated variant(s) to phenotype have been uncovered (*Lappalainen and MacArthur, 2021*; *Timpson et al., 2018*; *Visscher et al., 2017*). Here, we set out to investigate how particular noncoding regions harboring clustered variants associated with a common disease modulate expression of a disease-associated transcription factor gene in a tissue-specific manner, and how this expression change affects phenotype in vivo.

Genome-wide association studies (GWAS) have identified many common variants in over 100 genetic loci associated with atrial fibrillation (AF) risk, the most prevalent arrhythmia associated with high comorbidity and increased mortality risk (*Christophersen et al., 2017*; *Chugh et al., 2014*; *Low et al., 2017*; *Nielsen et al., 2018*; *Roselli et al., 2018*; *Staerk et al., 2017*). The identification of functional variants, REs, and target genes underlying AF will provide important insights into the molecular mechanisms of disease (*van Ouwerkerk et al., 2020*). AF-associated variants have been identified in loci harboring transcription factor-encoding genes, including *PITX2*, *TBX5*, and *PRRX1*, suggesting that altered expression levels of such factors cause imbalances in gene regulatory networks that control heart rhythm and function (*van Ouwerkerk et al., 2020*). Indeed, using mouse models, insufficiency of these transcriptional regulators was shown to cause arrhythmia susceptibility (*Bosada et al., 2021*; *Dai et al., 2019*; *Kirchhof et al., 2011*; *Laforest et al., 2019*; *Nadadur et al., 2016*; *Tao et al., 2014*; *Tucker et al., 2017*; *Wang et al., 2010*; *Zhang et al., 2019*). Heterozygous loss- or gain of function variants in *TBX5* can cause Holt-Oram syndrome in humans, characterized by congenital heart defects and cardiac conduction anomalies, as a result of profound changes in the gene regulatory networks controlling heart development and function (*Basson et al., 1997*; *Bruneau et al., 1999*; *Kathiriya et al., 2021*; *Li et al., 1997*; *Mori et al., 2006*; *Moskowitz et al., 2004*). Interestingly, duplications of TBX5 as well as intragenic duplications have been reported in families with (atypical) Holt-Oram syndrome including cardiac defects (*Cenni et al., 2021*; *Kimura et al., 2015*; *Patel et al., 2012*). Furthermore, a gain-of-function pathological missense variant in *TBX5* causes paroxysmal AF (*Postma et al., 2008*; *van Ouwerkerk et al., 2022*). Moreover, a previous study uncovered a 30% increase, rather than a reduction, in cardiac *TBX5* expression in human heart tissues has been associated with AF (*Roselli et al., 2018*). The effects of small but potentially physiologically relevant dosage increase in transcriptional regulators such as TBX5 are not well characterized.

We deleted the mouse orthologues of two AF variant-rich regions in the human *TBX5* locus to investigate how variant regions associated with a common disease modulate phenotype in a tissue- and developmental stage-specific manner. Each deletion caused a modest 30% increase in *Tbx5* expression in different heart compartments and at different stages of life. We report the relatively large effect of this modest increase in *Tbx5* expression on atrial function including arrhythmia susceptibility, and on the gene regulatory network. Decreased expression of *Prrx1* has been associated with AF in human and mouse models (*Roselli et al., 2018*; *Bosada et al., 2021*; *Tucker et al., 2017*). We observed a genetic interaction between *Tbx5* and *Prrx1*, and found that some of the transcriptional and functional changes in the atria caused by increased *Tbx5* expression were rescued by reducing cardiac *Prrx1* expression.

## Results

### Identification of two AF-associated regulatory regions in the *TBX5* locus

The topologically associated domain (*Dixon et al., 2012*; *Nora et al., 2012*) harboring *Tbx5* shows very limited contact with the adjacent domains harboring *Tbx3* and *Rbm19*, respectively (*van Weerd et al., 2014*). Because all AF-associated variants in this locus are found within the topologically associated domain of *TBX5* (*Figure 1A*), we anticipate that REs affected by the risk variants modulate the expression of *TBX5* only. Promoter capture Hi-C maps from iPSC-derived cardiomyocytes (*Montefiori et al., 2018*) show distinct contacts between the promoter of *Tbx5* and distal AF-associated regions, including the region in the last intron of *TBX5* (*Figure 1A*). To identify possible regulatory elements within the AF-associated regions, we analyzed epigenomic datasets in both human (*Gilsbach et al., 2018*; *van Ouwerkerk et al., 2019*) and mouse orthologous region (*Figure 1A–B*). We selected two regions: the first situated in the last intron (RE(int)), and the second immediately downstream (RE(down)) of *Tbx5*. Both fragments contain evolutionary conserved regions, and RE(int) harbors accessible chromatin sites in the left atria and ventricles, and EMERGE enhancer prediction signal (*van Duijvenboden et al., 2019*; *van Ouwerkerk et al., 2019*). Additionally, RE(int) and RE(down) contain regions associated with cardiac H3K4me1, and RE(int) with H3K27ac (*Gilsbach et al., 2014*; *Figure 1B*). The SNPs in the last intron and those downstream of the gene are clustered into two distinct haplotypes, suggesting that these two regions are independently inherited (*Figure 1—figure supplement 1*; *Machiela and Chanock, 2015*). Using CRISPR/Cas9 genome editing, we deleted these candidate REs from the mouse genome to test their function in vivo.

### *Tbx5* expression and arrhythmia predisposition are increased in atria of *Tbx5*[RE(int)KO] and *Tbx5*[RE(down)KO] mice

Before birth, expression of *Tbx5* in the atria was not different between genotypes (*Figure 2—figure supplement 1A*). However, in juvenile atria of RE(int) KO mice (p=0.001), *Tbx5* expression was slightly increased compared to controls (*Figure 2A*). Expression of *Rbm19*, *Tbx3*, and *Med13l*, which neighbor *Tbx5*, remained unchanged in atria or ventricles of both mutants (*Figure 2—figure supplement 1*). Both atria of *Tbx5*[RE(int)KO] adult mice expressed approximately 30% more *Tbx5*, whereas only the left atrium and lungs of *Tbx5*[RE(down)KO] adult mice expressed more *Tbx5* (*Figure 2B–C*). While we were not able to assess quantitative differences in protein expression, we observed that Tbx5 protein was selectively present in PCM-1 +cardiomyocyte nuclei in left and right atria of both control and of *Tbx5*[RE(int)KO] mice (*Figure 2—figure supplement 2*). We confirmed absence of aberrant splicing caused by the intronic deletion in *Tbx5*[RE(int)KO] (*Figure 2—figure supplement 3*). Interestingly, eQTL analysis indicated that AF is associated with a statistically significant 30% increase in *TBX5* expression in cardiac tissue (*Roselli et al., 2018*). This suggests that AF-associated variants in the corresponding human regions may mediate the increase in *TBX5* expression observed in patients with risk variants.

To identify functional AF-associated variants within the human intronic RE region, we tested enhancer activity of five fragments containing an AF-associated SNPs ($p<10^{-6}$) (*Roselli et al., 2018*) using luciferase assays in the atrial cardiomyocyte-like cell line HL-1 (*Figure 2D*). Of the tested fragments, the fragment containing rs7312625 showed increased enhancer activity, and the fragment containing rs4767237 showed minimal decreased activity (*Figure 2E*). Comparative TF motif analysis between the regions containing the major (A) or minor (G) allele of rs7312625 revealed that the minor (G) allele causes disruption of motifs for *Arabidopsis* SOL1/2 (TCX/TCX2) (*Supplementary file 1*), which are homologues of animal LIN54 (*Marceau et al., 2016*; *Simmons et al., 2019*), a DNA-binding component of the DREAM complex regulating cell cycle-dependent transcription (*Sadasivam and DeCaprio, 2013*). We also observed gain of a *sine oculis* (SIX) homeodomain transcription factor binding motif (*Meurer et al., 2021*; *Supplementary file 1*). Lin54 and both Six4 and Six5 are expressed in atrial tissue of mice (see *Supplementary files 2 and 3*).

Next, we recorded in vivo electrocardiograms (ECGs) to determine the functional consequences of such a modest increase in *Tbx5*. Both *Tbx5*[RE(int)KO] and *Tbx5*[RE(down)KO] mice had slower and more variable heart rates (R-R interval (RR); standard deviation of normal to normal R-R intervals (SDNN)) (*Figure 3A and B*). Additionally, we detected instances of sinus pauses and inverted P waves in mice with both deletions (*Figure 3—figure supplement 1A*). PR interval was significantly increased in *Tbx5*[RE(int)KO]

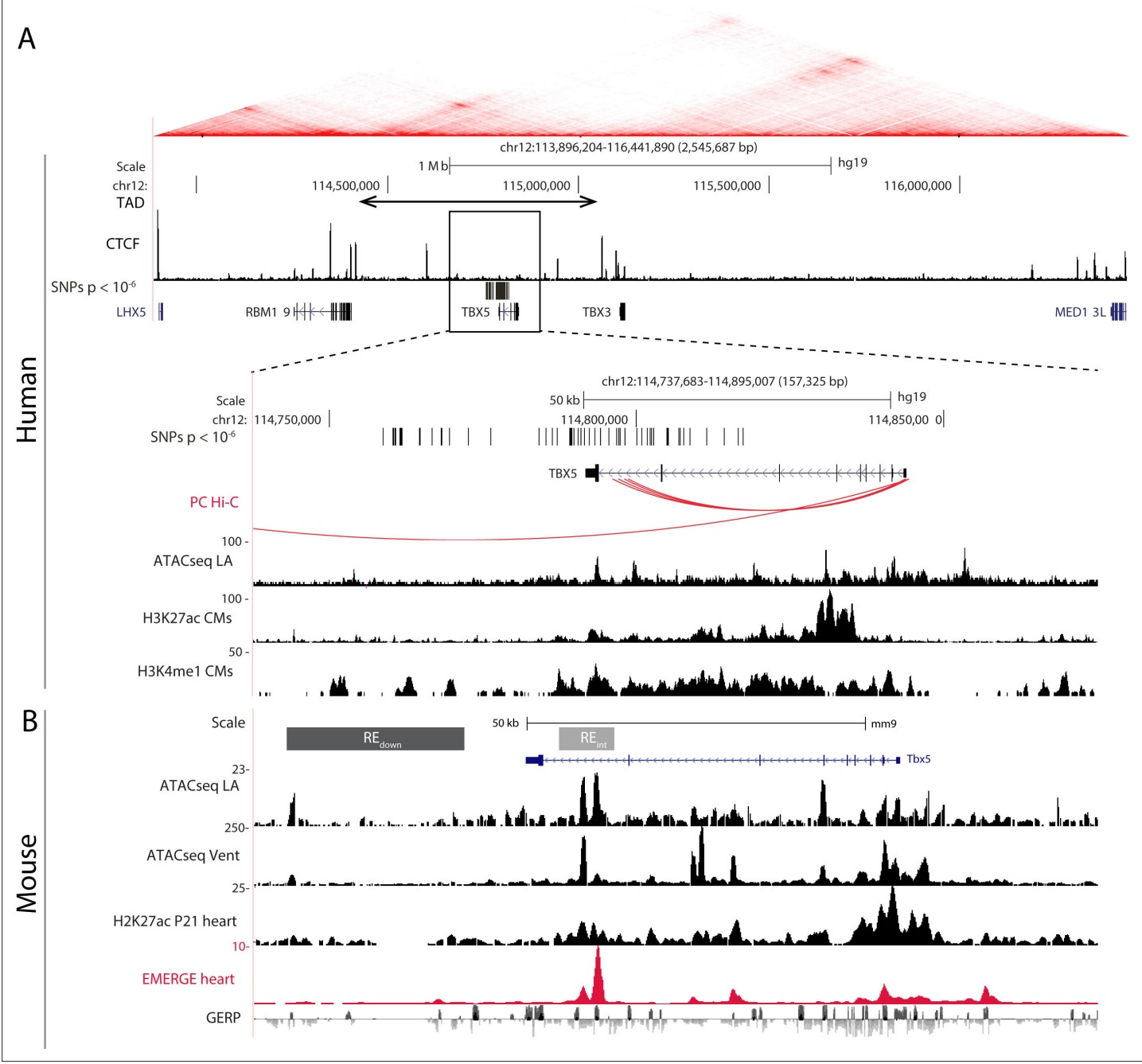

**Figure 1.** AF-associated noncoding variants are found in the *TBX5* locus. (**A**) Hi-C heatmap from human lymphoblastoid line GM12878 shows AF-associated variants are found in the regulatory domain of *TBX5*. Zoom-in of the AF-associated region overlaid with promoter capture Hi-C (red arcs), regions of open chromatin in whole left atria (ATACseq LA), H3K27ac and H3K4me1 ChIPseq signatures in cardiomyocytes. (**B**) Mouse orthologue of the human region including ATACseq from left atrial and ventricular CMs, H3K27ac ChIPseq from whole juvenile hearts, EMERGE, and conservation tracks. CRISPR/Cas9-generated deletions in light gray (REint), and dark gray (REdown).

The online version of this article includes the following figure supplement(s) for figure 1:

**Figure supplement 1.** AF-associated variants near *TBX5* are grouped into distinct haplotypes.

mice, but remained unaffected in $Tbx5^{RE(down)KO}$ mice (**Figure 3C**). Other ECG parameters were not affected (**Figure 3—figure supplement 1**). Heart rate corrected sinus node recovery time (cSNRT) and Wenckebach cycle length (WBCL) measured during transesophageal pacing (**Bosada et al., 2021**; **Verheule et al., 2004**) were increased in both mouse models (**Figure 3D and E**), suggesting altered

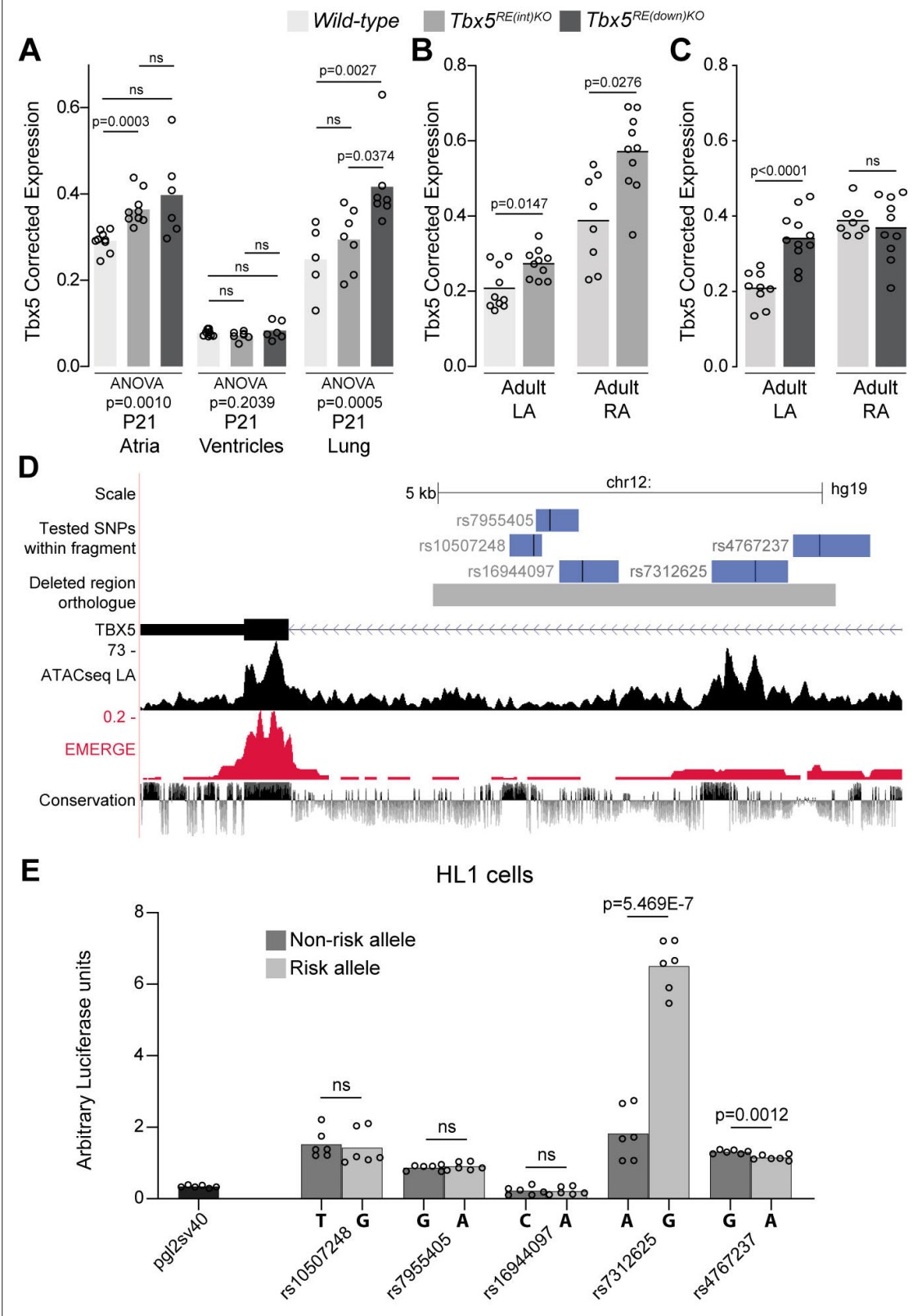

**Figure 2.** Deletion of AF-associated regions results in increased *Tbx5* in adult atrial tissue. (**A**) *Tbx5* expression in atria, ventricles, and lungs from P21 control, *Tbx5^{RE(int)KO}*, and *Tbx5^{RE(down)KO}* mice determined by RT-qPCR. (**B–C**) *Tbx5* expression levels in adult control and *Tbx5^{RE(int)KO}* (**B**) or *Tbx5^{RE(down)KO}* (**C**) left and right atria. (**D**) UCSC browser view of the human intronic region (gray) and tested fragments containing AF-associated SNPs (blue) overlaid with chromatin conformation, EMERGE, and conservation tracks. (**E**) Luciferase assay (n=6) shows enhancer activity differences between non-risk (dark

*Figure 2 continued on next page*

*Figure 2 continued*

gray) or risk (gray) alleles (Kruskal-Wallis p=0.0019). Statistical significance within each tissue type was determined with ANOVA followed by pairwise comparisons using Dunnett's T3 multiple comparison test in A, unpaired t-tests in B, C, and Kruskal-Wallis tests followed by pairwise unpaired t-tests (p values shown in figure) in E.

The online version of this article includes the following source data and figure supplement(s) for figure 2:

**Figure supplement 1.** No difference in transcript levels of genes adjacent to *Tbx5* in deletion mutants.

**Figure supplement 2.** Tbx5 is selectively expressed in cardiomyocyte nuclei in atria of *WT* and *Tbx5^{RE(int)KO}* mice.

**Figure supplement 3.** *Tbx5* splicing is not affected in deletion mutants.

**Figure supplement 3—source data 1.** Raw gel for *Figure 2—figure supplement 3*.

conduction system function. We next tested whether AA could be induced in both mouse lines using in vivo burst pacing (*Figure 3G* [typical AA traces]). The total duration of all AA episodes per mouse was greater in both deletion models when compared to control littermates (*Figure 3F* [top graph]). Yet, we were able to induce AAs more often in *Tbx5^{RE(int)KO}*, but not *Tbx5^{RE(down)KO}* mice compared to controls (*Figure 3F* [below graph]).

To further characterize the electrophysiological phenotypes of *Tbx5^{RE(int)KO}* mice we analyzed the properties of single isolated left atrial cardiomyocytes using patch-clamp. At 6 Hz stimulation, we observed APD prolongation in mutants at all measured repolarization stages, with no changes in AP upstroke velocity, resting membrane potential and maximal AP amplitude (*Figure 4A–C*). APD increase was present at all frequencies measured (*Figure 4D*). Because *Tbx5* has important roles in intracellular $Ca^{2+}$ handling (*Dai et al., 2019*; *Laforest et al., 2019*; *Zhu et al., 2008*), we measured intracellular $Ca^{2+}$ concentration ($[Ca^{2+}]_i$) in isolated atrial CMs using fluorescent calcium indicator Indo-1. We did not observe any changes in systolic or diastolic $[Ca^{2+}]_i$ concentration (*Figure 4E*). Together, our data show that a slight increase in *Tbx5* disturbs atrial function and can predispose to arrhythmia.

## Sensitivity of the atrial gene regulatory network to modestly increased *Tbx5* dosage

To gain insight into the mechanism underlying the changes in electrophysiological properties in mutants, we performed transcriptional profiling of whole left atria. We detected 13790 and 13564 different transcripts in *Tbx5^{RE(int)KO}* and *Tbx5^{RE(down)KO}*, respectively, of which 816 were significantly up- and 1235 were downregulated in *Tbx5^{RE(int)KO}* (*Figure 5A*; adjusted p<0.05), and 112 were up- and 123 were downregulated in *Tbx5^{RE(down)KO}* (*Figure 5B*; adjusted p<0.05). Both datasets shared a high proportion of deregulated genes, with fewer significantly deregulated genes observed in *Tbx5^{RE(down)KO}* compared to *Tbx5^{RE(int)KO}* (*Figure 5C*). These data indicate that transcriptomes in *Tbx5^{RE(int)KO}* atria and in *Tbx5^{RE(down)KO}* atria are comparably affected qualitatively, but not quantitatively, consistent with the slightly higher increase in *Tbx5* expression in *Tbx5^{RE(int)KO}* compared to *Tbx5^{RE(down)KO}*. Gene Ontology analysis (*Mi et al., 2019*) revealed that processes involved in cellular compartment organization, ion transport, and cardiac conduction characterized the transcripts found in the upregulated genes, and extracellular matrix organization, vasculature development, and actin cytoskeleton organization were found in the downregulated set in both deletions (*Figure 5D*). Accordingly, several *Tbx5* target genes known to affect APD or whose deregulation may result in electrophysiological changes, (*Dai et al., 2019*; *Laforest et al., 2019*; *Nadadur et al., 2016*; *Zhu et al., 2008*) were significantly deregulated in one or both deletion lines (*Figure 5F*, *Figure 5—figure supplement 1*).

Next, we compared the transcriptional response to increased *Tbx5* in left atria (*Tbx5^{RE(int)KO}*) to that of a gain-of-function missense mutation (*Tbx5^{G125R/+}*) (*van Ouwerkerk et al., 2022*), and noticed that it was unexpectedly divergent, with only 80 transcripts significantly deregulated in the same direction in both datasets (*Figure 5—figure supplement 2A*, *Supplementary file 4*). In contrast, comparison with left atria of inducible adult-specific *Tbx5* deletion (*Tbx5iKO*) (*Nadadur et al., 2016*) revealed that the majority of significantly deregulated genes common to both datasets were changed in opposite direction (*Figure 5—figure supplement 2B*, *Supplementary file 5*), thus indicating the *Tbx5*-dependent regulatory network responds in opposite direction to increased- or decreased *Tbx5* dosage, respectively. *PITX2* is strongly associated with AF and functionally implicated in atrial rhythm control (*Kirchhof et al., 2011*; *Tao et al., 2014*; *Wang et al., 2010*; *Zhang et al., 2019*), and was previously found to be a Tbx5 target in the left atrium (*Nadadur et al., 2016*). However, we did not

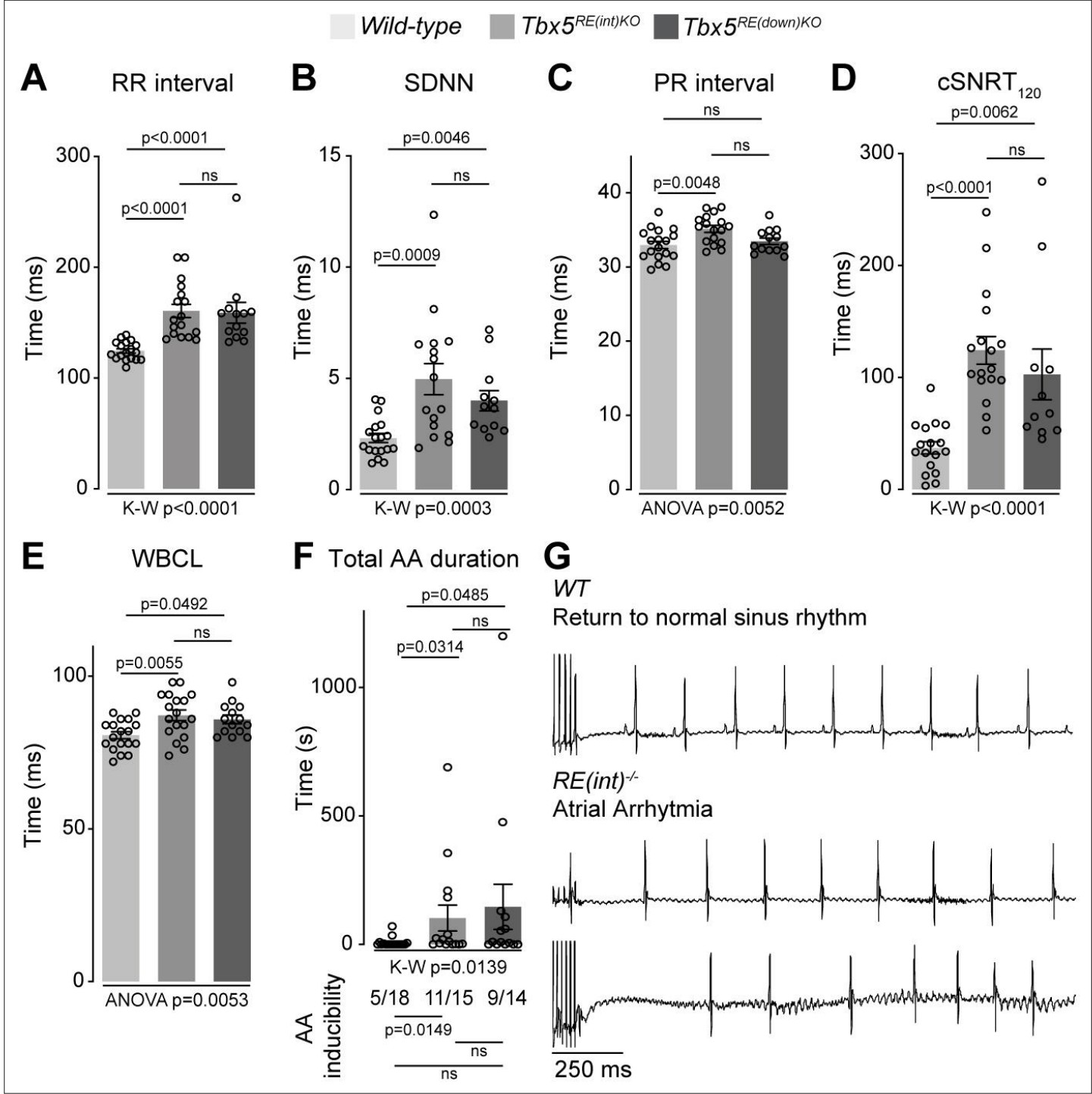

**Figure 3.** Increased *Tbx5* in adult atria results in altered in vivo electrophysiology. (**A–C**) Graphs show individual and average ECG measurements for heart rate (RR) (**A**), heart rate variation (SDNN) (**B**), and PR-interval (**C**) of adult *wild-type*, *Tbx5*$^{RE(int)KO}$, and *Tbx5*$^{RE(down)KO}$ mice. (**D, E**) Graphs show changes in heart rate-corrected sinus node recovery times at 120ms (cSNRT$_{120}$) (**D**), and Wenckebach cycle length (WBCL) (**E**). (**F**) Bar graph depicts the total time each mouse spent in an atrial arrhythmia (AA) episode after two pacing passes, with the number of mice in which at least one episode lasting >1 s was observed below each bar. (**G**) Representative traces from *wild-type* (top) and two *Tbx5*$^{RE(int)KO}$ (bottom) individuals showing disappearance of p waves or the start of atrial arrhythmia with variability in ventricular response after pacing stimulus. Significance for in vivo parameters was determined with Kruskal-Wallis test followed by Dunn's multiple comparison tests in A, B, D, and AA duration in F (top graph), and one-way ANOVA followed by Tukey's multiple comparisons test in C and E. AA inducibility significance was determined with pairwise Fisher's exact test (F bottom of graph).

The online version of this article includes the following figure supplement(s) for figure 3:

**Figure supplement 1.** Additional in vivo electrophysiology parameters.

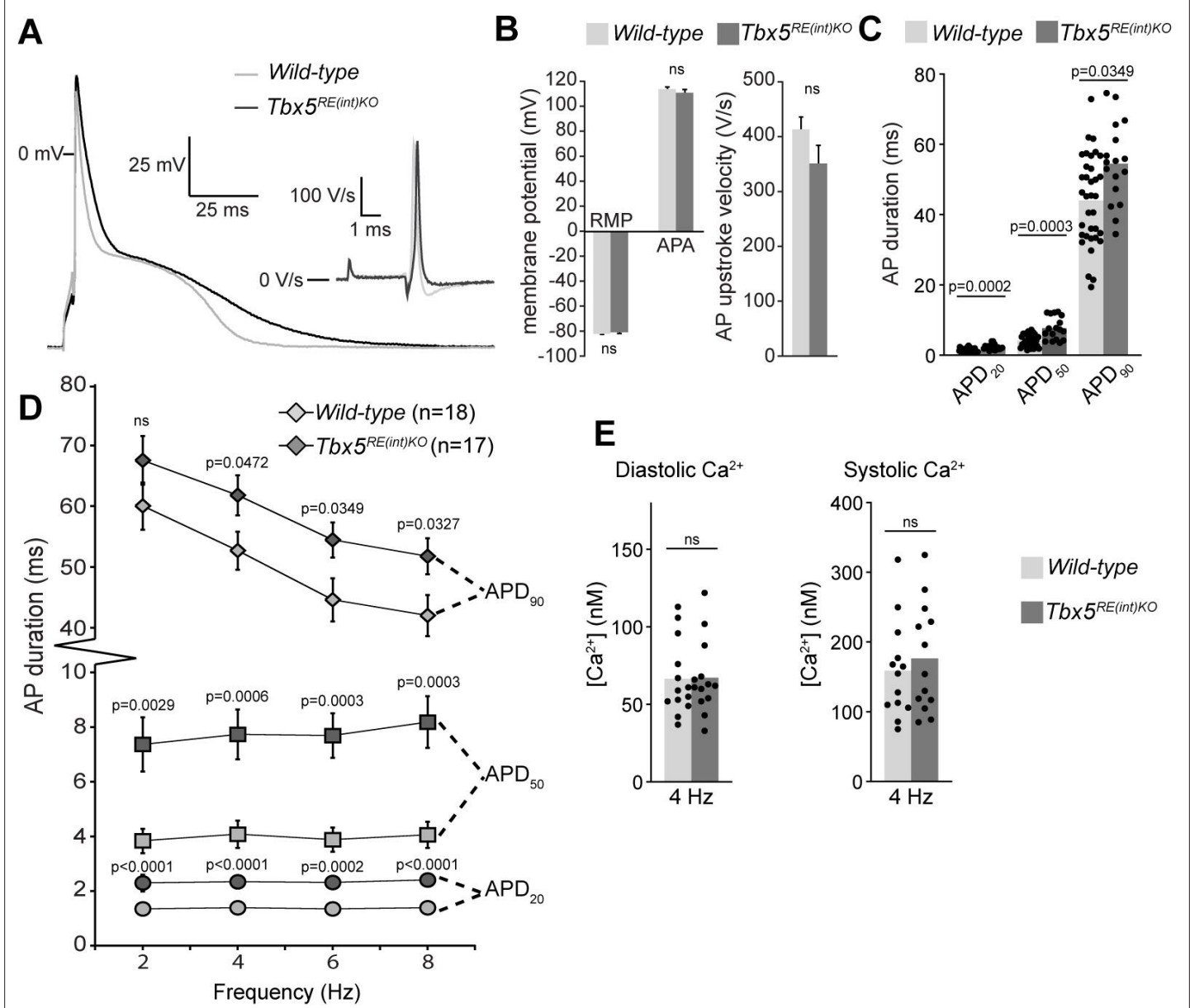

**Figure 4.** Atrial function abnormalities in $Tbx5^{RE(int)KO}$ mice. (**A**) Typical action potentials (APs) and upstroke velocity measured at 6 Hz in single left atrial cardiomyocytes using the amphotericin-perforated patch clamp technique. (**B**) Average resting membrane potential (RMP), AP amplitude (APA), and AP upstroke velocity remain unchanged in mutants. (**C**) AP duration at 20, 50, and 90% of repolarization ($APD_{20}$, $APD_{50}$, $APD_{90}$, respectively) is increased in mutants. (**D**) $APD_{90}$ was prolonged at all tested frequencies in mutant left atrial cardiomyocytes. Error bars are SD. (**E**) Diastolic and systolic intracellular $Ca^{2+}$ concentrations ($[Ca^{2+}]_i$) were not changed in $Tbx5^{RE(int)KO}$ compared to controls. Statistical significance in B was determined with unpaired t-tests with Welch's correction, and unpaired t-tests were used in B. Experimental groups were compared using two-way repeated measures ANOVA (**D**, **E**), followed by pairwise unpaired t-tests (**C**). Error bars are SEM.

detect a change in *Pitx2* expression in either the left or right atria of $Tbx5^{RE(int)KO}$ mice (*Figure 5—figure supplement 3*).

To determine whether a slight increase in *Tbx5* expression in cardiomyocytes results in changes in chromatin accessibility, we performed ATAC-sequencing profiling (ATACseq) of $Tbx5^{RE(int)KO}$ left atrial cardiomyocytes (*Buenrostro et al., 2015*). After peak-calling, we found a total of 85,569 accessible sites common to both genotypes, and only 15 sites of increased accessibility and 32 of decreased accessibility in mutants (*Figure 5—figure supplement 4A*). Two of the sites with decreased accessibility were found in the locus of *Tbx5* (one of which caused by the deletion of the intronic genomic

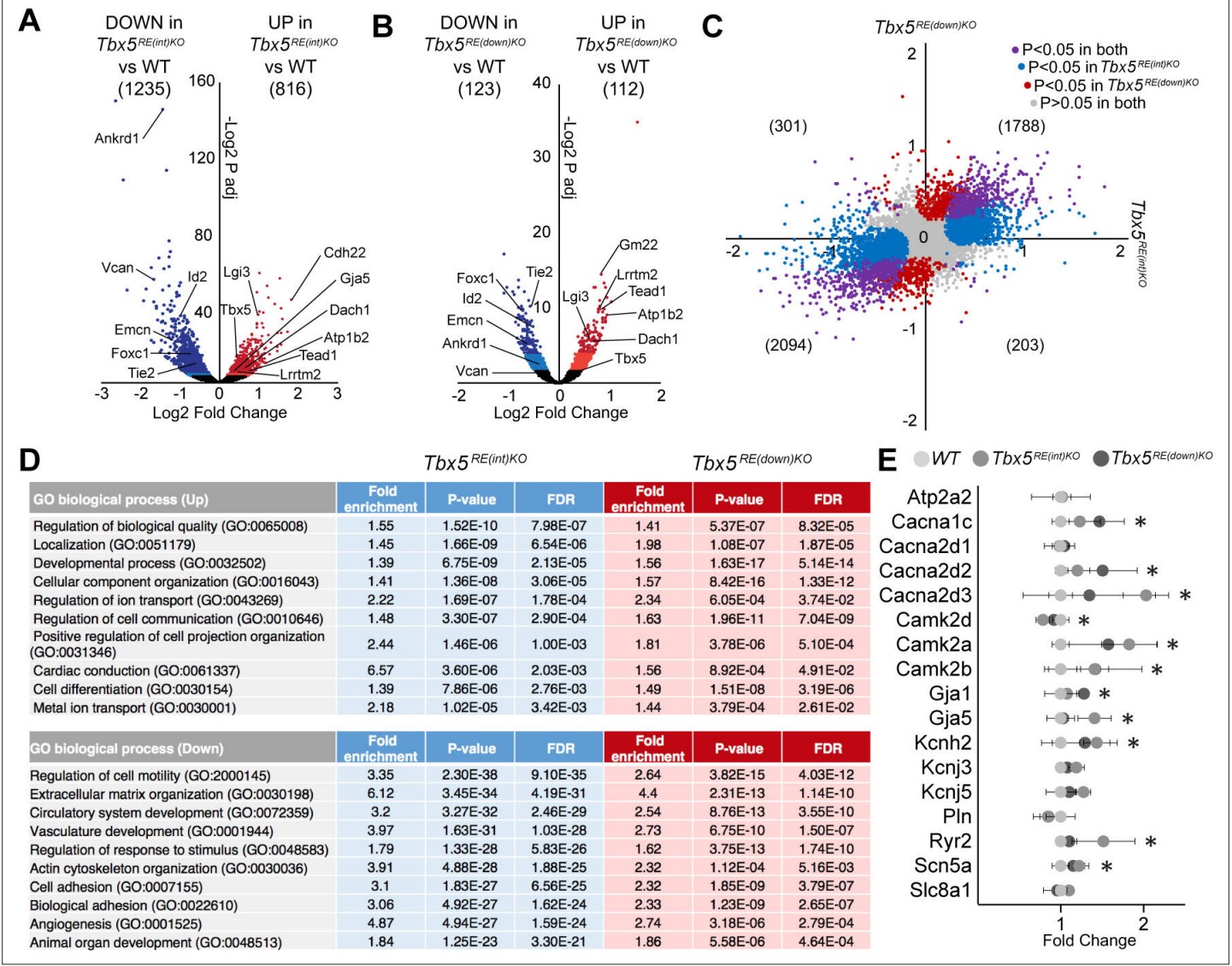

**Figure 5.** Transcriptomic analysis of $Tbx5^{RE(int)KO}$ left atria. (**A, B**) Volcano plot showing differentially expressed transcripts in $Tbx5^{RE(int)KO}$ (n=3)(**A**) and $Tbx5^{RE(down)KO}$ (n=3) (**B**) from *wild-type* (n=4) left atria. Dark red and light red dots indicate significantly upregulated genes by raw p-value and p-adjusted for multiple testing, respectively. Correspondingly, dark blue and light blue dots indicate downregulated genes. p Values were adjusted for multiple testing using the false discovery rate (FDR) method of Benjamini-Hochberg. (**C**) X-Y plot of all transcripts in $Tbx5^{RE(int)KO}$ (x axis) and $Tbx5^{RE(down)KO}$ (y axis), with deregulated genes common to both deletion mutants in purple, $Tbx5^{RE(int)KO}$ deregulated genes in blue, and $Tbx5^{RE(down)KO}$ deregulated genes in red. (**D**) Gene ontology (GO) analysis of upregulated and downregulated genes in $Tbx5^{RE(int)KO}$ and $Tbx5^{RE(down)KO}$ mutants. (**F**) Graph depicts fold change expression in control and mutant samples of genes known to affect action potential duration. * denotes significantly deregulated in one or both mutant lines.

The online version of this article includes the following figure supplement(s) for figure 5:

**Figure supplement 1.** Ion channel expression in left atria of deletion mutants.

**Figure supplement 2.** Comparison of genes differentially expressed between left atria of $Tbx5^{RE(int)KO}$ and control mice, $Tbx5^{G125R/+}$ and control mice and $Tbx5$ iKO and control mice.

**Figure supplement 3.** *Pitx2* expression is not affected in $Tbx5^{RE(int)KO}$ atria.

**Figure supplement 4.** Modest *Tbx5* upregulation minimally alters genome-wide chromatin accessibility.

---

fragment in $Tbx5^{RE(int)KO}$ mice), suggestive for increased direct transcriptional autoregulation in $Tbx5^{RE(int)KO}$ mice (**Figure 5—figure supplement 4B**). The very minor changes in chromatin accessibility indicate that a modest increase in *Tbx5* expression does not significantly change transcription factor occupancy or epigenetic state.

## An interaction between *Tbx5* and *Prrx1*, two AF-associated genes

We considered whether the presence of two AF-risk alleles would exacerbate the phenotype(s) consistent with AF. *PRRX1*, encoding the transcription factor Paired Related Homeobox 1, has been linked to AF-predisposition (*Bosada et al., 2021*; *Roselli et al., 2018*; *Tucker et al., 2017*). Reduced *PRRX1* expression in human cardiac tissues has been associated with AF (*Roselli et al., 2018*). We previously deleted the mouse orthologous variant region near *PRRX1* (Prrx1(enh)) to investigate its role in gene regulation and rhythm control (*Bosada et al., 2021*). *Prrx1*$^{(enh)KO}$ mice express less *Prrx1* specifically in cardiomyocytes compared to controls, and show atrial conduction slowing, lower AP upstroke velocity (indicative for lower Na$^+$ current densities; *Berecki et al., 2010*), as well as increased systolic and diastolic [Ca$^{2+}$]$_i$ concentration that culminate in increased susceptibility to atrial arrhythmia induction (*Bosada et al., 2021*). To explore whether and how these two AF-risk genes may interact, we intercrossed *Tbx5*$^{RE(int)KO}$ with *Prrx1*$^{(enh)KO}$ mice (decreased *Prrx1* in cardiomyocytes), and investigated cardiac transcriptomes and phenotypes across genotypes. We isolated cardiomyocyte and non-cardiomyocyte nuclei from whole hearts using a PCM-1 antibody and interrogated *Tbx5* and *Prrx1* expression across all genotypes. *Tbx5* was upregulated in *Tbx5*$^{RE(int)KO}$ cardiomyocytes, as expected, but not deregulated in *Prrx1*$^{(enh)KO}$ or double mutant cardiomyocytes (*Figure 6A*). *Prrx1* levels were increased in *Tbx5*$^{RE(int)KO}$ cardiomyocytes and decreased in *Prrx1*$^{(enh)KO}$, as expected, and also decreased in double mutant cardiomyocytes (*Figure 6B*). There were no statistically significant changes detected in the non-cardiomyocyte fractions (*Figure 6A and B*). These data indicate that in cardiomyocytes, Tbx5 regulates *Prrx1*, and that a regulatory feedback loop modulates *Tbx5* levels when *Prrx1* expression is reduced due to the Prrx1(enh) deletion.

We next performed transcriptome analysis and found that deregulated genes in left atria of *Tbx5*$^{RE(int)KO}$;*Prrx1*$^{(enh)KO}$ double mutants were more similar to the transcript changes found in *Tbx5*$^{RE(int)KO}$ than the ones found in *Prrx1*$^{(enh)KO}$, when comparing all lines to WT, suggesting a dominant contribution of *Tbx5*$^{RE(int)KO}$ to the transcriptional changes in double mutants (*Figure 6—figure supplement 1A-C*, *Supplementary files 6-7*). Using principal component analysis, we found that transcriptomes clustered according to genotype (*Figure 6C*). Compared to atrial transcriptomes of *Tbx5*$^{RE(int)KO}$ mice, those of *Tbx5*$^{RE(int)KO}$;*Prrx1*$^{(enh)KO}$ double homozygous showed less variance with WT atrial transcriptomes (*Figure 6C*, *Figure 6—figure supplement 1*, *Table 1*). This suggested an interaction between *Tbx5* and *Prrx1*, in which reduced *Prrx1* expression partially normalizes increased Tbx5-induced transcriptomic changes.

Next, we considered whether the adult electrophysiological changes observed in *Tbx5*$^{RE(int)KO}$ would also be rescued by the presence of this second AF-risk allele. In vivo ECGs and burst pacing experiments revealed that RR, PR interval, and WBCL were normalized in double homozygous mice, whereas SDNN, cSNRT, and AA inducibility remained unchanged (*Figure 6D–I*).

Taken together, our findings suggest that an interaction between *Tbx5* and *Prrx1* exists; in which *Prrx1* expression is induced by Tbx5, where *Prrx1* is required to induce *Tbx5* expression in *Prrx1*$^{(enh)KO}$ mice, and increased Tbx5 imposes changes in expression and electrophysiological properties, in part through increasing *Prrx1* expression in cardiomyocytes.

## Discussion

Our study reveals that mouse orthologues of two independent variant regions in the *TBX5* locus modulate *Tbx5* expression in tissue- and stage-specific manners, causing distinct specific phenotypes. While the impact of 50–100% dose reduction of *Tbx5* in humans and model systems has been well-investigated (*Bamshad et al., 1997*; *Bruneau et al., 2001*; *Kathiriya et al., 2021Li et al., 1997*; *Luna-Zurita et al., 2016*; *Mori et al., 2006*; *Moskowitz et al., 2004*), we here show the effect of physiologically relevant increases in expression of *Tbx5*. This is significant as both decreased and increased expression of genes has been associated with increased AF risk.(*Roselli et al., 2018*). Our study reveals that atrial- and postnatal-specific increase in *Tbx5* levels of only 30% affects postnatal atrial gene regulation, function and arrhythmia propensity. Furthermore, we provide an example of the interaction between the effect of two independent AF-associated variant regions (*TBX5* and *PRRX1*) on phenotype. Our models provide insight into the mechanisms underlying the pleiotropic effects and interactions of disease-associated regulatory variants, which typically cause small changes in target

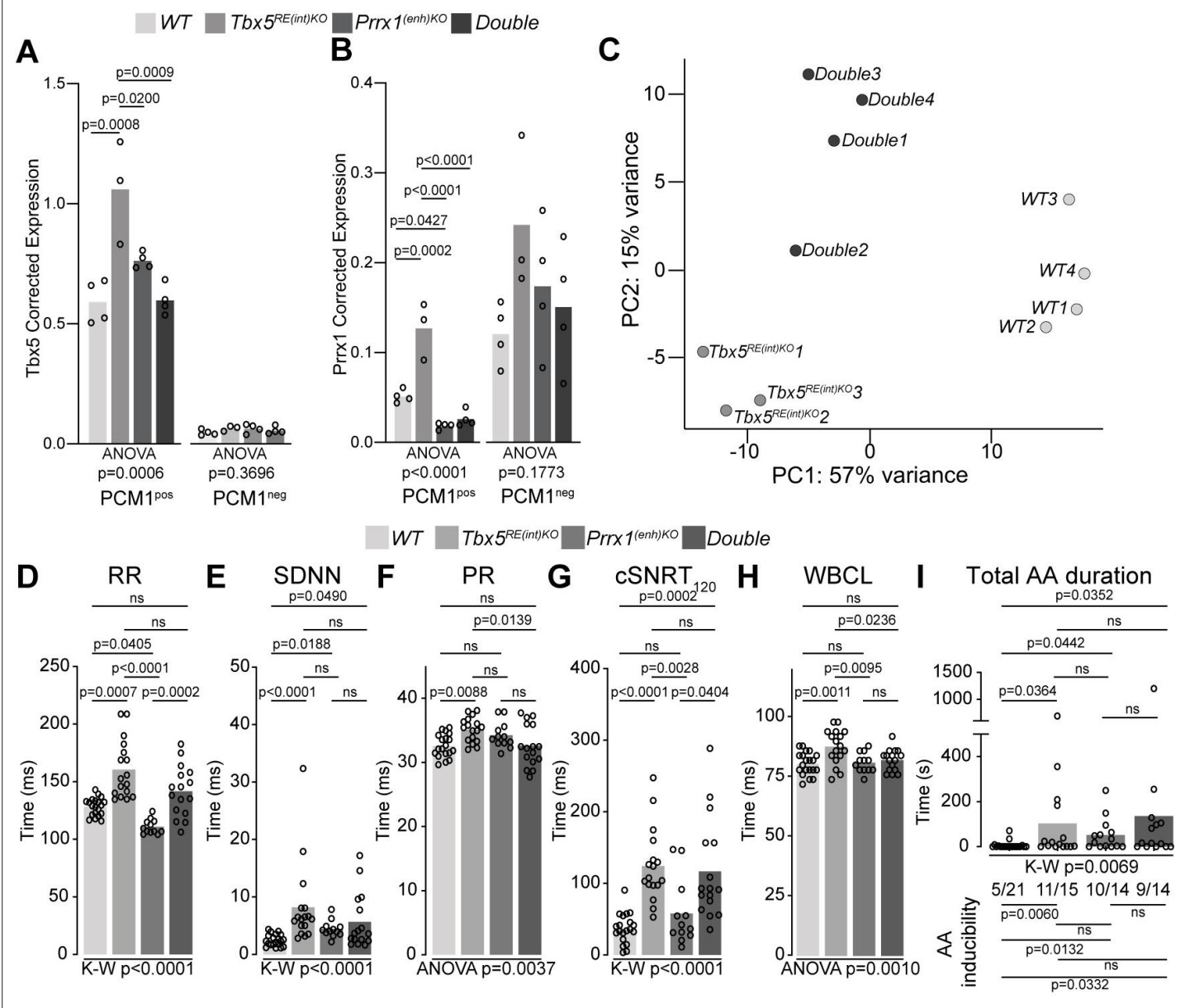

**Figure 6.** A genetic interaction between *Tbx5* and *Prrx1*. (**A, B**) Graph shows reference gene-corrected expression of *Tbx5* (**A**), and *Prrx1* (**B**) in PCM1-positive and -negative nuclei fractions of *wild-type*, *Tbx5^(RE(int)KO)*, *Prrx1^(enh)KO)* and *double* mutant whole hearts determined by RT-qPCR. (**C**) Principal component analysis of transcriptomes of *wild-type*, *Tbx5^(RE(int)KO)* and *Double*KO left atrial samples. (**D–H**) Graphs show individual and average ECG measurements for RR (**D**), heart rate variation (SDNN) (**E**), PR interval (**F**), cSNRT at 120ms pacing (**G**), and WBCL (**H**) of *wild-type*, *Tbx5^(RE(int)KO)*, *Prrx1^(enh)KO)* and *Double*KO mice. (**I**) Bar graph depicts the total time each mouse spent in an AA episode after two pacing passes, with the number of mice in which at least one episode lasting >1 s was observed below each bar. Statistical significance was determined using one-way ANOVA followed by Tukey's test for pairwise comparisons in A, B, F, H, Kruskal-Wallis test followed by Dunn's test for pairwise comparisons in D, E, G, and AA duration in I (top of graph), Fisher's exact test for pairwise comparisons of AA inducibility in I (bottom of graph).

The online version of this article includes the following figure supplement(s) for figure 6:

**Figure supplement 1.** Comparison of genes differentially expressed between left atria of *Prrx1^(enh)KO)* and control mice, *Tbx5^(RE(int)KO)* and control mice, and *Double homozygous* mutants and control mice.

gene expression in particular cell-types, conditions or stages of development (**GTEx Consortium, 2020**; **Sobreira et al., 2021**; **Strober et al., 2019**; **Watanabe et al., 2019**).

We found that the mouse orthologues of two noncoding regions studied here containing AF-associated variants in the *TBX5* locus harbor REs that independently controlled *Tbx5* expression in

**Table 1.** Gene ontology (GO) analysis of upregulated and downregulated genes in *Tbx5*[RE(int)KO], *Prrx1*[(enh)KO], and *Double homozygous* adult left atria.

| GO biological process | Prrx1(enh)KO | | | | Tbx5RE(int)KO | | | | Double | | | |
|---|---|---|---|---|---|---|---|---|---|---|---|---|
| | UP/DOWN | Fold enrich | P-value | FDR | UP/DOWN | Fold enrich | P-value | FDR | UP/DOWN | Fold enrich | P-value | FDR |
| Regulation of biological quality (GO:0065008) | DOWN | 1.64 | 1.7E-08 | 3E-06 | UP | 1.55 | 1.5E-10 | 8E-07 | UP | 1.56 | 3.4E-08 | 1.5E-05 |
| Localization (GO:0051179) | DOWN | 2.11 | 1.5E-14 | 1.6E-11 | UP | 1.45 | 1.7E-09 | 6.5E-06 | UP | 1.44 | 2.9E-07 | 8E-05 |
| Developmental process (GO:0032502) | UP | 2.33 | 6.1E-20 | 1.6E-16 | UP | 1.39 | 6.8E-09 | 2.1E-05 | UP | 1.61 | 1.8E-14 | 2.9E-10 |
| Cellular component organization (GO:0016043) | DOWN | 1.85 | 4.3E-08 | 6.9E-06 | UP | 1.41 | 1.4E-08 | 3.1E-05 | UP | 1.54 | 1.3E-10 | 2.2E-07 |
| Regulation of ion transport (GO:0043269) | DOWN | 2.68 | 8.7E-08 | 1.3E-05 | UP | 2.22 | 1.7E-07 | 0.00018 | UP | 2.36 | 7.6E-07 | 0.00018 |
| Regulation of cell communication (GO:0010646) | DOWN | 2.18 | 2.8E-18 | 1.1E-14 | UP | 1.48 | 3.3E-07 | 0.00029 | UP | 1.57 | 3.4E-07 | 8.9E-05 |
| Positive regulation of cell projection organization (GO:0031346) | DOWN | 2.07 | 0.00033 | 0.0158 | UP | 2.44 | 1.5E-06 | 0.001 | UP | 2.08 | 3.9E-07 | 0.0001 |
| Cardiac conduction (GO:0061337) | DOWN | 7.1 | 0.00011 | 0.00597 | UP | 6.57 | 3.6E-06 | 0.00203 | UP | 7.36 | 1E-05 | 0.0015 |
| Cell differentiation (GO:0030154) | UP | 2.25 | 2.8E-11 | 8.2E-09 | UP | 1.39 | 7.9E-06 | 0.00276 | UP | 1.65 | 4.8E-10 | 5.8E-07 |
| Metal ion transport (GO:0030001) | DOWN | 3.43 | 4.5E-08 | 7.3E-06 | UP | 2.18 | 1E-05 | 0.00342 | UP | 2.25 | 0.00069 | 0.049 |
| Regulation of cell motility (GO:2000145) | UP | 2.96 | 2.1E-14 | 1.2E-11 | DOWN | 3.35 | 2.3E-38 | 9.1E-35 | DOWN | 2.28 | 2.5E-05 | 0.00564 |
| Extracellular matrix organization (GO:0030198) | UP | 7.57 | 2E-24 | 3.1E-20 | DOWN | 6.12 | 3.5E-34 | 4.2E-31 | DOWN | 7.87 | 3.2E-17 | 5.1E-13 |
| Circulatory system development (GO:0072359) | UP | 3.3 | 2.8E-17 | 3.7E-14 | DOWN | 3.2 | 3.3E-32 | 2.5E-29 | DOWN | 3.14 | 2.3E-10 | 9.1E-07 |
| Vasculature development (GO:0001944) | UP | 4.11 | 4.6E-17 | 4.8E-14 | DOWN | 3.97 | 1.6E-31 | 1E-28 | DOWN | 3.23 | 3.2E-07 | 0.00017 |
| Regulation of response to stimulus (GO:0048583) | UP | 1.91 | 1.8E-17 | 2.6E-14 | DOWN | 1.79 | 1.3E-28 | 5.8E-26 | DOWN | 1.61 | 2.2E-06 | 0.00072 |

*Table 1 continued on next page*

*Table 1 continued*

| | Prrx1(enh)KO | | | Tbx5RE(int)KO | | | Double | | |
|---|---|---|---|---|---|---|---|---|---|
| Actin cytoskeleton organization (GO:0030036) | DOWN | 3.17 | 4.6E-08 | DOWN | 3.91 | 4.9E-28 | DOWN | 3.26 | 7.9E-07 |
| | | | 7.3E-06 | | | 1.9E-25 | | | 0.00034 |
| Cell adhesion (GO:0007155) | UP | 2.92 | 3.9E-12 | DOWN | 3.1 | 1.8E-27 | DOWN | 2.99 | 1.1E-08 |
| | | | 1.5E-09 | | | 6.6E-25 | | | 1.9E-05 |
| Biological adhesion (GO:0022610) | UP | 2.94 | 2E-12 | DOWN | 3.06 | 4.9E-27 | DOWN | 2.95 | 1.4E-08 |
| | | | 8.2E-10 | | | 1.6E-24 | | | 2E-05 |
| Angiogenesis (GO:0001525) | UP | 4.5 | 4.9E-12 | DOWN | 4.87 | 4.9E-27 | DOWN | 4.17 | 3.7E-07 |
| | | | 1.8E-09 | | | 1.6E-24 | | | 0.00019 |
| Animal organ development (GO:0048513) | UP | 2.03 | 8.4E-17 | DOWN | 1.84 | 1.3E-23 | DOWN | 1.78 | 3.4E-07 |
| | | | 7.4E-14 | | | 3.3E-21 | | | 0.00018 |

the heart in vivo. Deletion of either of the two RE-containing regions only affected *Tbx5* expression, consistent with the notion that *Tbx5* and surrounding RE regions are largely confined to one topologically associated domain not shared with adjacent genes (*van Weerd et al., 2014*). The mechanism of the repressive action of the REs and cross-talk with the other REs in the locus (*Smemo et al., 2012*) remain to be established. For RE(int), which showed the largest effect size in atria upon deletion, we identified two AF-associated SNPs that caused differential transcriptional activity of the RE sub fragment in HL1 atrial cardiomyocyte-like cells; one variant caused ~sixfold increase and the other minimally decreased activity. The risk allele of the first variant, rs7312625, disrupts a potential Lin54 motif, implying loss of interaction with the DREAM complex (*Sadasivam and DeCaprio, 2013*), and thus suggesting possible cell cycle-dependent regulation that remains to be elucidated. We speculate that in individuals carrying the variant that increases the activity of RE(int) in HL-1 cells, the repressive activity of RE(int) is reduced in the context of the entire regulatory system, thus causing increased *TBX5* expression. The opposite could be true for the second variant, yet the small effect would likely be experimentally undetectable. On the other hand, the AF-associated variants in RE(down) may increase *Tbx5* expression in both the left atrium and pulmonary vein myocardium. The pulmonary veins have been strongly implicated in AF as they are the most common source of triggered activity (*Haïssaguerre et al., 1998*; *Jaïs et al., 1997*). *TBX5* levels in pulmonary vein myocardium may influence gene regulation independently from the atrial *TBX5* levels (*Steimle et al., 2020*). The variants in RE(down) segregate independently from those in RE(int), indicating these variant RE regions act through distinct tissue-specific transcriptional mechanisms that are influenced by AF-predisposing common variants. A further implication is they may cumulatively increase AF predisposition in a manner dependent on risk variant dose, in which homozygous carriers of risk haplotypes in RE(int) and RE(down) have the largest relative predisposition. These relations could be addressed in future genotype-phenotype analyses of human atrial and pulmonary vein samples.

Heart rates of the $Tbx5^{RE(int)KO}$ mice were slower and more variable than of controls. In addition, PR interval was prolonged in $Tbx5^{RE(int)KO}$ mice combination with increased Wenckebach cycle lengths. The latter is in line with the important role of Tbx5 in sinus node and AV node development and function (*Mori et al., 2006*; *Moskowitz et al., 2007*; *Moskowitz et al., 2004*; *van Eif et al., 2018*). Moreover, atrial arrhythmias could be more easily induced in $Tbx5^{RE(int)KO}$ mice suggesting electrophysiological remodeling of the atria. It has been shown that mice haploinsufficient for *Tbx5* show atrial downregulation of genes encoding proteins involved in cardiac conduction (e.g. *Gja1*/Cx43 and *Scn5a*/Nav1.5) and $Ca^{2+}$ handling (e.g *Atp2a2*/Serca2a and *Ryr2*/Ryr2) (*Nadadur et al., 2016*; *Zhu et al., 2008*). This leads to slower conduction, reduced $Ca^{2+}$ concentration in the sarcoplasmic reticulum $[Ca^{2+}]_{SR}$, increased incidence of early and delayed afterdepolarization, and accordingly, increased AF propensity (*Laforest et al., 2019*; *Nadadur et al., 2016*). In the atria of our $Tbx5^{RE(int)KO}$ mice, where expression of *Tbx5* was slightly elevated, transcriptomic analysis indicated upregulation of genes involved in cardiac conduction including *Scn5a* and *Cx43*. The $Ca^{2+}$ handling genes *Atp2a2*/Serca2a and *Ryr2*/Ryr2 were not differently expressed between $Tbx5^{RE(int)KO}$ mice and controls. Accordingly, in our mice, we did not observe any changes in $Ca^{2+}$ transients and long decay constants as seen in *Tbx5* haploinsufficient mice (*Laforest et al., 2019*). Expression of *Cacna1c* and *Cacna2d2*/*Cacna2d3*, encoding subunits of the L-type $Ca^{2+}$ channel, was upregulated in the atrium of $Tbx5^{RE(int)KO}$ mice. This is consistent with the observed longer atrial APD in $Tbx5^{RE(int)KO}$ mice compared to controls, which may predispose to increased arrhythmia inducibility, as seen in mice harboring the pathogenic TBX5-G125R variant or *Prrx1* enhancer deletion (*Bosada et al., 2021*; *van Ouwerkerk et al., 2022*). Although shortened APD is typically found in AF patients with sustained arrhythmia (*Kim et al., 2002*; *Wu et al., 2001*), prolonged APD has been previously associated with increased risk of developing AF (*Bosada et al., 2021*; *Lee et al., 2016*; *Olson et al., 2006*; *Simpson et al., 1988*; *van Ouwerkerk et al., 2022*).

The molecular mechanisms underlying electrophysiological remodeling have remained unclear. For example, atrial cardiomyocytes of *Tbx5* iKO (*Tbx5* deletion induced in adult mice), of $Tbx5^{+/G125R}$ (heterozygous for a gain of function missense mutation) and of $Tbx5^{RE(int)KO}$ mice show APD prolongation (*Nadadur et al., 2016*; *van Ouwerkerk et al., 2022*). When comparing the transcriptional changes in left atria of these mouse models, we observed that it was unexpectedly divergent between the $Tbx5^{RE(int)KO}$ and $Tbx5^{+/G125R}$ models, whereas significantly differentially expressed genes common to both $Tbx5^{RE(int)KO}$ and *Tbx5* iKO models were changed in opposite direction (*Figure 5—figure*

*supplement 2*, *Supplementary files 4 and 5*). This suggests that the degree and direction of change of expression of, for example, genes implicated in APD (including *Cacna2d3*, *Camk2a*, *Kcnh2*, and *Ryr2*) do not necessarily predict changes in APD. Furthermore, *Pitx2* was downregulated in left atria of *Tbx5* iKO mice and upregulated in right atria of *Tbx5*$^{+/G125R}$ mice (*van Ouwerkerk et al., 2022*), and Tbx5 and Pitx2 were found to work antagonistically in the left atrium to tightly regulate the expression of genes impacting on cardiac electrophysiology (*Nadadur et al., 2016*). Yet, we did not detect a change in *Pitx2* expression in the atria of *Tbx5*$^{RE(int)KO}$ mice. Together, these findings indicate that decreasing or increasing Tbx5 dose or changing Tbx5 function induces a large number of divergent transcriptional responses that disturb the balance in the genetic networks underlying functional (electrophysiological) properties of the atria, such as APD, leading to similar phenotypic outcomes.

The risk of developing complex diseases such as AF is strongly influenced by a large number of pleiotropic variants that each confers a small change in overall risk. We found that Tbx5 drives *Prrx1* expression, and that *Tbx5* levels are Prrx1-dependent (or Prrx1 target-dependent) in specific contexts. We then asked whether combining two alleles modeling the changes in gene expression conferred by AF-associated variant regions would cause a greater effect on heart phenotype, and by extension, arrhythmia predisposition, than each allele alone. Here, we found that introduction of this AF-susceptibility allele *Tbx5*$^{RE(int)KO}$ mice rescued heart rate, PR interval, and atrioventricular node phenotypes. This example shows that two variant regions that are both independently associated with increased risk for a particular disease, do not necessarily cumulatively increase disease predisposing phenotype, but may neutralize each other's effect. Previously, the atrial electrophysiological phenotype of *Tbx5* haploinsufficient mice was observed to be rescued by haploinsufficiency of either *Pitx2* or *Gata4*, both of which have been associated with AF as well (*Laforest et al., 2019*; *Nadadur et al., 2016*). Although the direction of change of expression of the transcription factors in these models does not necessarily correspond to the direction of change in AF patients, this highlights the inherent robustness of the underlying gene regulatory networks, which in general remain stable when individual quantitative parameters such as transcription factor dose or binding sites affinity change (*Albergante et al., 2014*). This also implies that evaluating specific interactions between AF risk loci will be necessary for ascertaining individual risk from genetic association data.

In conclusion, our data provide unique mechanistic insights into the biological effects of variant region-driven modest physiologically relevant changes in expression of crucial transcriptional regulators, and into the impact of interactions between transcriptional output-modulating risk loci on atrial biology.

## Materials and methods

### Usage of epigenomic datasets

The following publicly available epigenomic datasets were used: Hi-C (*Rao et al., 2014*), AF variants (*Roselli et al., 2018*), promoter capture Hi-C maps from iPSC-derived cardiomyocytes (*Montefiori et al., 2018*), accessible chromatin in human and mouse left atria (*van Ouwerkerk et al., 2019*), cardiac H3K27ac and H3K4me1 ChIP-seq signatures in human (*Gilsbach et al., 2014*), and EMERGE enhancer prediction signal (*van Duijvenboden et al., 2019*; *van Ouwerkerk et al., 2019*).

### Generation of mutant mice

Mutant mice were generated using CRISPR/Cas9. Guide RNA (sgRNA) constructs were designed with ZiFiT Targeter (*Sander et al., 2010*). The sgRNA constructs were transcribed in vitro using MEGAshortscript T7 (Invitrogen AM1354) and mMessage Machine T7 transcription kit (Invitrogen AM1344) according to manufacturer instructions. One-cell FVB/NRj zygotes were microinjected with 10 ng/µL each sgRNA and 25 ng/µL Cas9 mRNA to generate mouse founders. Deletions were validated by PCR and Sanger sequencing. The sgRNA target sequences are the following: RE(int) guide 1 (GGGAAATC GCCTTACCTTTC), guide 2 (GGACTGTTGGGTCACCTTGT), and RE(down) guide 1 (GGCTCCTTCGTC AGTAAATA), guide 2 (AAACAAGGGCTCTCTGGCGTTT). The deleted coordinates are the following in mm10: RE(int) chr5:119,874,013–119,880,148, and RE(down) chr5:119,891,017–119,915,744. Founders were backcrossed with wildtype FVB/NJ mice to obtain stable lines. Downstream experiments were performed on F3-F7 mice, backcrossed with wild-type FVB mice. All transgenic mice were

maintained on a FVB/NJ background commercially obtained from Jackson laboratory (stock number 100800).

For tissue harvest, animals were euthanized by 20% $CO_2$ inhalation followed by cervical dislocation.

## EdU cardiomyocyte proliferation assay

Timed pregnant female mice received intraperitoneal injection of 100 mg/kg EdU one hour before sacrifice by isoflurane and cervical dislocation. E14.5 fetuses were isolated in 1xPBS on ice, the head was removed and the body fixed in 4% PFA for 24 hr. 7 µm sections were stained with mouse-anti-actin (1:400; Sigma A9357), goat-anti-nkx2.5 (1:150; Santa Cruz sc-14033), DAPI (1:1000) and EdU Click-IT (ThermoFisher, C10340) before imaging and counting. Nkx2.5 positive nuclei were designated cardiomyocytes, Nkx2.5 and EdU double positive nuclei were designated proliferating cardiomyocytes. Ratios were normalized within each litter.

## qPCR

Total RNA was isolated from atria, ventricle and lungs from P21 and left and right atria from adult male and female mice using ReliaPrep RNA Tissue Miniprep System (Promega, Z6112) according to the manufacturer's protocol. cDNA was reverse transcribed with oligo dT primers from 500 ng of total RNA, or random hexamers from 500 pg of CM nuclear RNA, according to the manufacturer's protocol of the Superscript II Reverse Transcriptase system (Thermo Fisher Scientific, 18064014). Expression levels of candidate target genes were determined by quantitative real-time PCR using a LightCycler 480 Instrument II (Roche Life Science, 05015243001). Expression levels were measured using LightCycler 480 SYBR Green I Master (Roche, 04887352001) and the primers had a concentration of 1 pmol/L. The amplification protocol consisted of 5 min 95 °C followed by 45 cycles of 10 s 95 °C, 20 s 60 °C and 20 s 72 °C. Relative start concentration (N0) was calculated using LinRegPCR (*Ruijter et al., 2009*). Values were normalized to the geometric mean of two reference genes per experiment (Hprt, Ppia, or Rpl32) (*Ruiz-Villalba et al., 2017*). The primer sequences are as follows: (all 3' to 5') Hprt: TGTTGGATATGCCCTTGACT, GATTCAACTTGCGCTCATCT; Ppia: GGGTGGTGACTTTACA CGCC, CTTGCCATCCAGCCATTCAG; Rpl32: GCCTCTGGTGAAGCCCAAG, TTGTTGCTCCCATAAC CGATGT; Tbx5: CCCGGAGACAGCTTTTATCG, TGGTTGGAGGTGACTTTGTG; Prrx1: CACAAGCA GACGAAAGTGTGG, GTTGTCCTGTTTCTCCGCTG; Tbx3: CGCCGTTACTGCCTATCAGAA; GCCA TTGCCAGTGTCTCGAA.

## Cell culture and transfection luciferase assays

RE sub fragments were cloned into a modified pGL2-Basic plasmid containing an SV40 minimal promoter and an adjusted multiple cloning site for in vitro analysis by a transfection luciferase assay. HL1 cells (RRID:CVCL_0303) were grown in 24-well plates in Claycomb medium (Sigma-Aldrich, 51,800 C) supplemented with chemically defined HL-1 FBS substitute (Lonza, 77227), Glutamax (ThermoFisher Scientific, 35050–061) and Pen/Strep (ThermoFisher Scientific, 15070–063). Cells were transfected using polyethylenimine 25 kDa (PEI, Brunschwig, 23966–2) at a 1:3 ratio (DNA:PEI). Standard transfections were carried out using 200 ng of reporter construct per well. 24 hr after transfection, cells were lysed using Renilla luciferase assay lysis buffer (Promega, E291A-C) and luciferase activity measured. Luciferase measurements were performed using a GloMax Explorer (Promega, GM3500). During the measurement, 100 µL D-Luciferin (p.j.k, 102111) was injected (150 µL/s) followed by a 1-s delay and 5 s of measurement. Transfections were carried out at least three times and measured in duplicates. HL-1 cell cultures were routinely tested negative for mycoplasma contamination. HL-1 cell lines were easily distinguished based on cellular morphology and contractility. HL-1 is not found in the database of commonly misidentified cell lines that is maintained by the International Cell Line Authentication Committee.

## In vivo electrophysiology

12–20 week old male mice were anesthetized with 5% Isoflurane (Pharmachemie B.V. 061756) and placed on thermostated mat (36 °C) with a steady flow of 1.5% isoflurane during all experiments. Electrodes were inserted subcutaneously in the limbs and connected to an ECG amplifier (Powerlab 26T, AD Instruments). The electrocardiogram (ECG) was measured for 5 min. ECG parameters were determined in Lead II (L-R) based on the last 60 s of the recording. For atrial stimulation, an octapolar

CIB'er electrode (NuMED) was advanced through the esophagus to achieve atrial capture. Atrial capture thresholds were determined for each mouse, and all pacing protocols were performed at 2 x threshold. For sinus node recovery time (SNRT) measurements, a 4-s pacing train with a cycle length of 120 or 100ms was used. SNRT was defined as the interval between the last pacing stimulus and onset of the first P wave. To control for differences in sinus rate, SNRT was normalized to resting heart rate (cSNRT = SNRT – RR interval). To determine the Wenckebach cycle length (WBCL) we applied a 4s pacing train starting at a cycle length of 100ms, and decreasing by 2ms until atrioventricular (AV) block was first observed. Atrial arrhythmia (AA) was induced by 1 or 2 s bursts starting with a cycle length of 60ms, decreasing successively with a 2 ms decrement, down to a cycle length of 10ms. AA duration was the sum of time each mouse spent under and AA episode after completion of the two passes. Atrial arrhythmia (AA) inducibility was scored as the number of mice in which at least one episode lasting >1 s of arrhythmia was induced after pacing. Mice of both sexes were used for atrial arrhythmia induction experiments.

## Cellular electrophysiology

Single cells were isolated from left atria of adult male mice by enzymatic dissociation. Therefore, excised hearts were perfused for 5 min in a Langendorff system with a modified Tyrode's solution containing (in mmol/L): NaCl 140, KCl 5.4, $CaCl_2$ 1.8, $MgCl_2$ 1.0, glucose 5.5, HEPES 5.0; pH 7.4 (set with NaOH). Subsequently, the hearts were perfused with Tyrode's solution containing a low $Ca^{2+}$-concentration (10 μmol/L) for 10 min, after which Liberase TM research grade (Roche Diagnostics, GmbH, Mannheim, Germany, 5401119001) and Elastase from porcine pancreas (Bio-Connect B.V., Huissen, Netherlands, W59168R) were added for 12 min at a concentration of 0.038 mg/mL and 0.01 mg/mL, respectively. All solutions were saturated with 100% $O_2$ and the temperature was maintained at 37 °C. To obtain single cells, the digested left atria was cut into small pieces which were triturated for 4 min through a pipette (tip diameter: 0.8 mm) in the low $Ca^{2+}$ Tyrode's solution, supplemented with 10 mg/ml Bovine Serum Albumin (Roche Diagnostics, essential fatty free, fraction V). Single cells were stored at room temperature for at least 45 min before they were used. Quiescent single rod-shaped cells with smooth surfaces were selected for electrophysiological measurements.

Action potentials (APs) were recorded with the amphotericin-B perforated patch-clamp technique using an Axopatch 200B amplifier (Molecular Devices Corporation, Sunnyvale, CA). Voltage control and data acquisition were performed as described previously (*Kamel et al., 2021*), and APs were low-pass filtered at 5 kHz and digitized at 40 kHz. Potentials were corrected for the estimated liquid junction potentials (*Barry and Lynch, 1991*).

APs were recorded at 36 ± 0.2°C using the modified Tyrode's solution. Pipettes (borosilicate glass (Harvard Apparatus, UK)) were filled with solution containing (in mmol/l): K-gluc 125, KCl 20, NaCl 5, amphotericin-B 0.44, HEPES 10, pH 7.2 (KOH). APs were elicited at 2–8 Hz by 3ms,~1.2 × threshold current pulses through the patch pipette. We analyzed resting membrane potential (RMP), maximal AP amplitude (APA), maximum AP upstroke velocity, and AP duration at 20, 50, and 90% repolarization ($APD_{20}$, $APD_{50}$, and $APD_{90}$, respectively). Parameters from 10 consecutive APs were averaged.

## Intracellular $Ca^{2+}$ measurements

Intracellular calcium concentrations: $[Ca^{2+}]_i$ were measured at 25 °C in HEPES solution ((mmol/l): $Na^+$ 156, $K^+$ 4.7, $Ca^{2+}$ 1.3, $Mg^{2+}$ 2.0, $Cl^-$ 150.6, $HCO_3$ 4.3, $HPO_2^-$ 1.4, HEPES 17, Glucose 11 and 1% fatty acid free albumin, pH 7.3) using the fluorescent probe Indo-1 as described previously (*Verkerk et al., 2003*). In brief, isolated myocytes were exposed to 5 μmol/l of the acetoxymethyl esters of indo-1 during 30 min at 37 °C. Myocytes were attached to a poly-D-lysine (0.1 g/l) treated cover slip placed on a temperature-controlled microscope stage of an inverted fluorescence microscope (Nikon Diaphot) with quartz optics. A temperature-controlled perfusion chamber (height 0.4 mm, diameter 10 mm, volume 30 μL, temperature 37 °C), with two needles at opposite sides for perfusion purposes, was tightly positioned over the cover slip. The contents of the chamber could be replaced within 100ms. Bipolar square pulses for field stimulation (40 V/cm) were applied through two thin parallel platinum electrodes at a distance of 8 mm. One quiescent single myocyte was selected (myocytes with more than one spontaneous oscillation per 10 s were excluded) and the measuring area was adjusted to the cell surface with a rectangular diaphragm. The wavelength of excitation of Indo-1 was 340 nm, applied with a stabilized xenon-arc lamp (100 W). Fluorescence was measured in dual emission mode at 410

and 516 nm. Emitted light passed a barrier filter of 400 nm, a dichroic mirror (450 nm) and respective narrow band interference filters in front of two photomultipliers (Hamamatsu R-2949). Signals were digitized at 1 kHz and corrected for background signals recorded from Indo-1 free myocytes. Ten subsequent Ca$^{2+}$ transients were averaged from which apparent [Ca$^{2+}$]$_i$ was calculated according to the ratio equation (*Grynkiewicz et al., 1985*).

## Isolation of CM nuclei

Nuclei isolation was performed as follows: snap frozen adult left and right atria from adult male and female mice were trimmed and homogenized in lysis buffer containing RNase inhibitor using an Ultra-Turrax homogenizer. Samples were further homogenized with a loose pestle douncer (10 strokes). After a 10 min incubation in the lysis buffer, an additional 10 strokes were performed with a tight pestle. The lysis procedure was monitored by light microscopy to ensure complete tissue and cell lysis and efficient nuclear extraction. The crude lysate was successively passed through 100 and 30 µm mesh filters. The final lysate was spun at 1000 × g for 5 min and the resulting pellet was resuspended in 500 µl staining buffer (5% BSA in PBS) supplemented with RNAse inhibitor. Isolated nuclei were incubated with rabbit polyclonal antibodies specific for pericentriolar material 1 (PCM1) (Sigma-Aldrich; HPA023370) at a dilution of 1:400 for 1 hr rotating at 4 °C. Next, Alexa Fluor 647-conjugated donkey-anti-rabbit 647 antibodies (ThermoFisher Scientific A-31573; 1:500 dilution), and DAPI (1:1000 dilution) were added and the incubation was continued for an additional hour. Samples were spun at 1000 × g for 10 min and washed with 500 µl staining buffer before resuspension in 500 µl staining buffer supplemented with RNAse inhibitor. Intact CM nuclei were sorted on a BD Influx FACS on the basis of DAPI and Alexa Fluor 647 positivity into cold BL +TG buffer from the ReliaPrep RNA Tissue Miniprep System (Promega, Z6112) for RNA isolation, and into resuspension buffer for ATACseq (*Buenrostro et al., 2015*). RNA was isolated following a gDNA depletion step according to the manufacturer's instructions. RNA yield and purity was assessed using an Agilent 2100 Bioanalyzer in combination with the RNA Pico chips.

## Library preparation and sequencing

For RNA isolated from left atria, 500 ng was used for library generation with the KAPA mRNA Hyper-Prep kit (Roche) and sequenced on the HiSeq4000 system (Illumina) with 50 bp single-end reads. RNAseq sample sizes are as follows: Whole adult left atria: 4 WT, 3 *Tbx5*$^{RE(int)KO}$, 3 *Tbx5*$^{RE(down)KO}$, 4 *Prrx1*$^{(enh)KO}$, and 3 *Double homozygous*.

## Differential expression analysis

Reads were mapped to the mm10 build of the mouse transcriptome using STAR (*Dobin et al., 2013*). Differential expression analysis was performed using the DESeq2 package based on a negative binomial distribution model (*Love et al., 2014*). p-Values were corrected for multiple testing using the false discovery rate (FDR) method of Benjamini-Hochberg. We have used 0.05 as FDR control level. Unsupervised hierarchical clustering was performed on differentially expressed genes using the R package pheatmap version 1.0.8. (http://cran.rproject.org/web/packages/pheatmap/index.html). PANTHER (*Mi et al., 2019*) was used for gene ontology (GO) biological process analysis. Benjamini–Hochberg correction was performed for multiple testing-controlled p values. Statistically significant enriched terms were functionally grouped and visualized.

## ATACseq on CM nuclei

ATACseq on FACS-sorted PCM1 +nuclei was performed and analyzed as described in *Buenrostro et al., 2015*. Approximately 50 k nuclei were used as input. The library was sequenced (paired-end 125 bp) and data was collected on a HiSeq4000.

## Peak-calling and motif analysis of ATACseq

Reads from ATAC-seq data were mapped to mm10 build of the mouse genome using BWA (*Li and Durbin, 2009*), the default settings were used. The BEDTools suite was used to distribute the genome wide signal into bins of 500 bp (*Quinlan and Hall, 2010*). Bins with less than 89 cumulative tags across all 10 samples were discarded as noise. Differential accessibility was assessed using the DESeq2 package based on a model using the negative binomial distribution (*Love et al., 2014*). p-Values were

corrected for multiple testing using the false discovery rate (FDR) method of Benjamini-Hochberg. We have used 0.05 as FDR control level. Continuous bins with differential signal were subsequently merged together using the BEDTools suite.

200 bp summits were determined of ATAC-seq performed in atrial cells (*Fernandez-Perez et al., 2019*). In total, HOMER (*Heinz et al., 2010*) was performed on sequences of 2000 neutral called peaks (randomly sampled out of 85,570 peaks), 2000 peaks up (randomly sampled out of 3551 peaks), and 1776 peaks down in *Tbx5*$^{RE(int)KO}$, all with random genome as background in HOMER.

Unsupervised hierarchical clustering was performed on differentially detected peaks and genes in RNA-seq and ATAC-seq using the R package pheatmap, version 1.0.8. (http://cran.r-project.org/web/packages/pheatmap/index.html).

### Usage of EMERGE
The genome-wide heart enhancer prediction track generated by EMERGE (*van Duijvenboden et al., 2016*) was used as a proxy for the presence of putative heart enhancers. The particular prediction used aimed at identifying robustly active heart enhancers. This was accomplished by further annotating the available true positives (*Visel et al., 2007*) to train the algorithm with on the basis of the consistency in heart activity patterns shown. Full details are outlined in *van Ouwerkerk et al., 2019*.

### Statistics
The experimenters were blind to mouse genotype during all measurements and outcome assessment. Datasets were tested for normality using Shapiro-Wilk test unless specified otherwise. Whole tissue RT-qPCR fold change vs WT in fetal ventricles and adult whole left atria corrected expression changes were analyzed with unpaired t-tests and reference gene-corrected *Tbx5* in juvenile whole tissue was analyzed with Welch's ANOVA followed by Dunnet's T3 multiple comparison tests within each tissue type. Luciferase assays on HL1 cells were analyzed using Kruskal-Wallis test. In vivo electrophysiology was analyzed with Kruskal-Wallis followed by Dunn's multiple comparison tests or one-way ANOVA followed by Tukey's multiple comparison tests. Significance of atrial arrhythmia (AA) duration was determined with Mann Whitney Wilcoxon test and differences in AA inducibility were tested using Fisher's exact test. Conduction velocity was analyzed with unpaired t-tests with Welch's correction. For single cell and whole tissue action potential (AP) duration (APD) normality and equal variance assumptions were tested with the Kolmogorov-Smirnov and the Levene median test, respectively. Two groups were compared with unpaired t-test or repeated measures ANOVA followed by pairwise comparison using the Student-Newman-Keuls test. Differences in $Ca^{2+}$ transient amplitude as well as changes in diastolic and systolic $Ca^{2+}$ concentrations in atria were tested using two-way ANOVA. Multiple testing corrections were performed independently within each hypothesis. Data are presented as individual data points and mean or mean ± standard error of the mean (SEM) or standard deviation (SD), as indicated, and <0.05 defines statistical significance. Statistical analysis was performed using GraphPad Prism 9.

## Acknowledgements
We thank Berend de Jonge for technical assistance. This work was supported by CardioVasculair Onderzoek Nederland (CVON) project 2014–18 CONCOR-genes Young Talent Program (to FMB), CVON project 2014–18 CONCOR-genes (to VMC), Leducq Foundation 14CVD01 (to VMC), Dutch CardioVascular Alliance OUTREACH (to VMC) and ZonMW TOP 91217061 (to VMC).

## Additional information

### Funding

| Funder | Grant reference number | Author |
| --- | --- | --- |
| CardioVasculair Onderzoek Nederland | Project 2014-18 CONCOR-genes Young Talent Program | Fernanda M Bosada |

| Funder | Grant reference number | Author |
|---|---|---|
| Fondation Leducq | 14CVD01 | Vincent M Christoffels |
| Dutch Cardiovascular Alliance | OUTREACH | Vincent M Christoffels |
| ZonMw | TOP 91217061 | Vincent M Christoffels |
| Dutch Research Council | OCENW.GROOT.2019.029 | Vincent M Christoffels |

The funders had no role in study design, data collection and interpretation, or the decision to submit the work for publication.

## Author contributions

Fernanda M Bosada, Conceptualization, Data curation, Formal analysis, Investigation, Visualization, Writing – original draft, Writing – review and editing; Karel van Duijvenboden, Data curation, Visualization, Writing – review and editing; Alexandra E Giovou, Formal analysis, Investigation, Methodology; Mathilde R Rivaud, Formal analysis, Writing – review and editing; Jae-Sun Uhm, Arie O Verkerk, Data curation, Formal analysis, Writing – review and editing; Bastiaan J Boukens, Data curation, Formal analysis, Supervision, Investigation, Methodology, Writing – original draft, Writing – review and editing; Vincent M Christoffels, Conceptualization, Supervision, Funding acquisition, Writing – original draft, Writing – review and editing

## Author ORCIDs

Fernanda M Bosada http://orcid.org/0000-0001-9817-7370
Karel van Duijvenboden http://orcid.org/0000-0001-5691-8097
Mathilde R Rivaud http://orcid.org/0000-0002-0680-3483
Arie O Verkerk http://orcid.org/0000-0003-2140-834X
Bastiaan J Boukens http://orcid.org/0000-0001-6449-145X
Vincent M Christoffels http://orcid.org/0000-0003-4131-2636

## Ethics

Housing, husbandry, all animal care and experimental protocols were in accordance with guidelines 310 from the Directive 2010/63/EU of the European Parliament and Dutch government. Protocols were 311 approved by the Animal Experimental Committee of the Amsterdam University Medical Centers. 312 Animal group sizes were determined based on previous experience.

## Decision letter and Author response

Decision letter https://doi.org/10.7554/eLife.80317.sa1
Author response https://doi.org/10.7554/eLife.80317.sa2

# Additional files

## Supplementary files

- Supplementary file 1. rs7312625 motif analysis +- 30 bp. Changed motifs in red.
- Supplementary file 2. DESeq2 result WT vs $Tbx5^{RE(int)KO}$ adult left atria.
- Supplementary file 3. DESeq2 result WT vs $Tbx5^{RE(down)KO}$ adult left atria.
- Supplementary file 4. RNAseq comparison of $Tbx5^{RE(down)KO}$ and $Tbx5^{G125R/+}$ adult left atria.
- Supplementary file 5. RNAseq comparison of $Tbx5^{RE(down)KO}$ and $Tbx5iKO$ adult left atria.
- Supplementary file 6. DESeq2 result WT vs Double adult left atria.
- Supplementary file 7. DESeq2 result WT v $Prrx1^{(enh)KO}$ adult left atria.
- Supplementary file 8. Deletion coordinates in mm10.
- MDAR checklist

## Data availability

Adult left atrial RNAseq and ATACseq have been deposited under GEO accession numbers GSE189342 and GSE189498.

The following datasets were generated:

| Author(s) | Year | Dataset title | Dataset URL | Database and Identifier |
|---|---|---|---|---|
| Bosada FM | 2022 | Tbx5 regulatory element controls atrial function and heart size | https://www.ncbi.nlm.nih.gov/geo/query/acc.cgi?acc=GSE189342 | NCBI Gene Expression Omnibus, GSE189342 |
| Bosada FM | 2022 | Tbx5 regulatory element controls atrial function and heart size | https://www.ncbi.nlm.nih.gov/geo/query/acc.cgi?acc=GSE189498 | NCBI Gene Expression Omnibus, GSE189498 |

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
