## [Editor Report]

Molecular mechanisms of atrial fibrillation, a highly prevalent arrhythmia, have been challenging to contextualize. The investigators, in a series of experiments using the deletion of two loci in TBX5 (one intronic and one downstream), achieved a modest increase in Tbx5 in the atria. In a series of elegant experiments they show that this increase in TbX5 in turn impacted the cardiac gene regulatory network that resulted in a higher susceptibility to AF. These results will provide useful insights into the mechanisms of disease especially the variable phenotypes observed with functional variants of a gene.

---

## [Decision Letter]

**Decision letter after peer review:**

Thank you for submitting your article "An atrial fibrillation-associated regulatory region modulates cardiac *Tbx5* levels and arrhythmia susceptibility" for consideration by *eLife*. Your article has been reviewed by 3 peer reviewers, including Kalyanam Shivkumar as the Reviewing Editor and Reviewer #1, and the evaluation has been overseen by a Senior Editor. The following individual involved in the review of your submission has agreed to reveal their identity: Benoit Bruneau (Reviewer #2).

Essential revisions:

Please address the discussion points raised by reviewers # 1 and 2.

1. Please provide data for this comment by reviewer # 2-Does the increase in Tbx5 gene expression result in an increase in protein expression?

2. Please address the extra data sought by reviewer # 3 in the detailed comments.

*Reviewer #1 (Recommendations for the authors):*

1. Why is the assumption that the regulatory elements "modulate the expression of TBX5 only" as opposed to have trans effects on distant gene encoding regions? For example, in reference 41 on the topic of promoter capture Hi-C maps, "Remarkably, more than 90% of SNP-target gene interactions did not involve the nearest gene, while 40% of SNPs interacted with at least two genes, demonstrating the importance of considering long-range chromatin interactions when interpreting functional targets of disease loci." Aside from the two regions for selected to study, were additional regions identified?

2. As "13790 and 13564 different transcripts in RE(int)-/- and RE (down)-/-" were identified, how is it known that the transcriptional changes are from increased TBX5 levels as opposed to a direct, trans effects due to the deletion of the two loci at TBX5? This seems at odds with the claim that "[d]eletion of either of the two RE-containing regions only affected Tbx5 expression".

3. As the SNP of interest rs7312625 has been identified as a variant associated with the electrocardiographic P wave (https://pubmed.ncbi.nlm.nih.gov/28794112/) and the transgenic mice in this study had slower heart rates, sinus pauses, inverted P waves (presumably ectopic atrial rhythm), PR prolongation, corrected sinus node recovery time, and Wenckebach cycle length, could the findings be more explanatory for nodal (sinus and atrioventricular) function and the cardiac conduction system more generally rather than atrial fibrillation in particular? The study seems as though it could have been re-framed as an evaluation of nodal function rather than atrial fibrillation. Of note, these findings were of note in the RE(down)-/- mice, which did not have increased atrial arrhythmia inducibility.

4. The authors cite reference 19 as showing a "slight increase…in cardiac TBX5 expression in human heart tissues [as being] associated with AF". What was the magnitude of the increase? Challenging to gather this information on this reviewer's review of reference 19.

5. Of all the genes that are modulated by changes in TBX5 expression, why was Prrx1 chosen? How do the authors explain that lack of change noted for Pitx2?

*Reviewer #2 (Recommendations for the authors):*

The rationale for focusing on RE(down) is not clearly appreciated based on Figure 1. This is in contrast to the apparent evidence in support of RE(int). Promoter Hi-C data shows DNA interactions in human that appears to span beyond the presumptive syntenic region of RE(down) in mouse, as well as a lack of chromatin accessibility, histone marks or EMERGE prediction for RE(down). Perhaps highlighting relevant features for RE(down) more explicitly would address this minor concern.

For supplemental figure 1, please define LD as "linkage disequilibrium" prior to abbreviation, as appropriate, and list the abbreviation at the beginning of the manuscript. As well, a scale bar is not apparent.

Does the increase in Tbx5 gene expression result in an increase in protein expression?

It would be interesting to compare these mouse models to the recently published TBX5 gain of function model.

For Figure 3, please define RR as "R-R interval" in legend and SDNN as "standard deviation of normal to normal R-R intervals", as appropriate, and add to abbreviation list.

In Figure 5, where is Prrx1 highlighted in the transcriptomic analysis?

For Figure 6C, how does Prrx1(enh)-/- compare in the PCA analysis? Please display it.

*Reviewer #3 (Recommendations for the authors):*

1. TBX5 mutations in humans and animal models are known to generate congenital heart defects and upper limb/radial malformations. Perhaps of relevance to this manuscript, human patients with TBX5 duplications also show many of these same phenotypes (Heinritz et al. 2005 Heart; Patel et al. 2012 Eur J Hum Genet; Kimura et al. 2015 Pedatr Cardiol; Cenni et al. 2021 Eur J Hum Genet). Do RE(int) and RE(down) homozygous mutations result in congenital heart or limb malformations? Structural heart defects can certainly impact cardiac electrophysiology and reporting structural defects, or lack of them can help put the phenotype into context.

2. The RE(int) deletion appears to be more severe than the RE(down) deletion. Does the RE(int) deletion interfere with the splicing of the last two exons? Depending on library construction, perhaps the RNA-seq can address this.

3. In Figure 2E, the authors report that rs7312625 A>G appears to be the functional SNP as it alone results in increased Tbx5 expression in vitro. As this is a predicted regulatory element, what transcription factor binding motif(s) is this SNP predicted to interfere or generate? Does this motif change make sense with gained Tbx5 expression in cardiomyocytes/HL1 cells?

4. In Figure 3C, the authors report RE(int) mutants have an increased PR interval and in Figure 4C, the authors report RE(int) have an increased APD20, 50, and 90. Single-copy loss of Tbx5 in mice has been reported to cause these same phenotypes (e.g. Bruneau et al. 2001 Cell; Nadadur et al. 2016 Sci Trans Med). Would you not expect the opposite phenotype, i.e., decreased PR interval and APD20/50/90? What is the explanation for these paradoxical results?

5. Several transcriptional profiling datasets have been generated for various Tbx5 mutations in mouse and human including a few by the authors (by cursory search of GEO for "TBX5"). How do the gene expression changes seen in the current datasets compare with previously generated datasets? Are the changes largely consistent with expectations of increased Tbx5 levels in RE(int) and RE(down)? It appears most public datasets are in Tbx5/TBX5 loss-of-function contexts, so we are expecting the opposite results. Does this data support that hypothesis or is there something else going on?

6. Figures 2A and 2C report that RE(down) demonstrated increased expression of Tbx5 in the adult left atria and P21 lungs, specifically, but there is little to no mention of this in the text or discussion. Given the importance of the pulmonary veins in the development of atrial fibrillation and prior publication history from the authors and colleagues on the topic of left atrial Tbx5, can the authors expand their discussion on the RE(down) related SNPs? The SNPs associated with RE(down) are independently segregating and the differences observed in this manuscript may hint at an independent, tissue-dependent function for that regulatory element.

---

## [Author Response]

Essential revisions:Please address the discussion points raised by reviewers # 1 and 2.1. Please provide data for this comment by reviewer # 2-Does the increase in Tbx5 gene expression result in an increase in protein expression?2. Please address the extra data sought by reviewer # 3 in the detailed comments.

We would like to thank the reviewers for their constructive criticism and suggestions. In our revised manuscript, we have addressed all comments and questions raised as detailed below. We believe that the incorporated changes have greatly improved the manuscript.

Reviewer #1 (Recommendations for the authors):1. Why is the assumption that the regulatory elements "modulate the expression of TBX5 only" as opposed to have trans effects on distant gene encoding regions? For example, in reference 41 on the topic of promoter capture Hi-C maps, "Remarkably, more than 90% of SNP-target gene interactions did not involve the nearest gene, while 40% of SNPs interacted with at least two genes, demonstrating the importance of considering long-range chromatin interactions when interpreting functional targets of disease loci." Aside from the two regions for selected to study, were additional regions identified?

Although a regulatory element, potentially containing SNP(s), usually shares the topologically associating domain (TAD) with its target genes, it does not necessarily interact with the nearest gene. In the referred study, promoter capture Hi-C identified only a fraction of the interactions, and showed a bias against detecting proximal interactions because of the high background signal associated with genomic locations close to each other. Moreover, the interactions reflect conformation (3D structure) of the chromatin and do not necessarily imply functional interactions. Nevertheless, our group has previously shown that the *Tbx5* regulatory region is confined to its own TAD (Weerd et al., 2014). Additionally, we examined the expression of genes adjacent to the Tbx5 TAD, *Rbm19*, *Tbx3* and *Med13l,* in atria and ventricles of mice with deletions by RT-qPCR, and found no changes in their expression (Figure 2—figure supplement 1). These data confirm that the deleted regulatory regions target *Tbx5* and not genes in adjacent TADs.

We identified at least one additional AF-associated variant region with regulatory potential in the *TBX5/Tbx5* locus, namely, in the second to last intron, which can be seen in Figure 1. However, we chose not to study this region because of the decreased accessibility in atrial tissue (ATACseq) and the comparably lower degree of conservation across species (GERP).

2. As "13790 and 13564 different transcripts in RE(int)-/- and RE (down)-/-" were identified, how is it known that the transcriptional changes are from increased TBX5 levels as opposed to a direct, trans effects due to the deletion of the two loci at TBX5? This seems at odds with the claim that "[d]eletion of either of the two RE-containing regions only affected Tbx5 expression".

Although we cannot fully rule out that the deleted regulatory regions target other genes in addition to *Tbx5*, we anticipate that the transcriptional changes are a result of increased *Tbx5* levels rather than a consequence of the deletion of the variant regions. As mentioned above in response to comment 1 of the reviewer, previous research indicated that the regulatory elements of *Tbx5* are restricted to the *Tbx5* TAD (Weerd et al., 2014), and we confirmed that nearby genes in adjacent TADs were not influenced by the deletions (see Figure 2—figure supplement 1 in the revised manuscript). Regulatory element-target promoter interactions are generally assumed to be confined to the same TAD (Dixon et al., 2012; Nora et al., 2012; Sexton et al., 2012), and it is considered very unlikely that these interactions span multiple TADs or chromosomes. Furthermore, we observed the upregulation of several well established TBX5 target genes (e.g. *Gja5*, *Ryr2*), suggesting that increased *Tbx5* dose is causal. Finally, we compared the transcriptomes of left atria of RE(int)^-/-^ mice (deletion in the last intron of *Tbx5*; increased *Tbx5* expression in atria) to the reported transcriptome of left atria of mice in which *Tbx5* was inactivated (Nadadur et al., 2016). We observed a strikingly opposite transcriptional response (see new Figure 5—figure supplement 2), indicating that the Tbx5-dependent regulatory network responds in opposite direction to increased and decreased Tbx5 dose, respectively.

3. As the SNP of interest rs7312625 has been identified as a variant associated with the electrocardiographic P wave (https://pubmed.ncbi.nlm.nih.gov/28794112/) and the transgenic mice in this study had slower heart rates, sinus pauses, inverted P waves (presumably ectopic atrial rhythm), PR prolongation, corrected sinus node recovery time, and Wenckebach cycle length, could the findings be more explanatory for nodal (sinus and atrioventricular) function and the cardiac conduction system more generally rather than atrial fibrillation in particular? The study seems as though it could have been re-framed as an evaluation of nodal function rather than atrial fibrillation. Of note, these findings were of note in the RE(down)-/- mice, which did not have increased atrial arrhythmia inducibility.

We thank the reviewer for this comment. The study the reviewer refers to states: “The P wave on an ECG is a measure of atrial electric function, and its characteristics may serve as predictors for atrial arrhythmias.” Therefore, the SNPs identified in this study may impact on expression of genes, including *TBX5* and *SCN5A*, involved in atrial cardiomyocyte electrophysiology. We addressed this aspect here, and found that atrial cardiomyocytes showed altered electrophysiological properties facilitating arrhythmia. The observed inverted P waves, changed SAN recovery times and changed WCL indeed suggest that SAN and AVN function may be affected in addition. Therefore, it is possible that the atrial arrhythmia is a culmination of multiple triggers, including changes in nodal and working cardiomyocytes. While we certainly want to further investigate a role of Tbx5 in SAN function, we feel that further characterization of nodal function remains outside the scope of this study.

4. The authors cite reference 19 as showing a "slight increase…in cardiac TBX5 expression in human heart tissues [as being] associated with AF". What was the magnitude of the increase? Challenging to gather this information on this reviewer's review of reference 19.

We thank the reviewer for pointing this out. We have amended the text in the revised manuscript to clarify a 30% increase in cardiac *TBX5* expression in lines 96-98 and 144-145.

5. Of all the genes that are modulated by changes in TBX5 expression, why was Prrx1 chosen?

In GWAS studies, the *PRRX1* locus has been strongly associated with AF. *PRRX1* encodes a transcription factor that we and others have shown to be causally linked to atrial fibrillation susceptibility. Therefore, we chose to intercross RE(int)^-/-^ and Prrx1(enh)^-/-^ mice to test whether the two established AF-risk alleles would aggravated the arrhythmogenic phenotype. Strikingly, we found a partial rescue of the phenotypes instead. We later determined that *Prrx1* was upregulated in RE(int)^-/-^ specifically in cardiomyocytes, but not in bulk RNAseq, likely because *Prrx1* is most highly expressed in non-cardiomyocyte cell populations.

We have changed the revised manuscript to clarify this in 229-2243, as follows:

“We considered whether the presence of two AF-risk alleles would exacerbate the phenotype(s) consistent with AF. *PRRX1*, encoding the transcription factor Paired Related Homeobox 1, has been linked to AF-predisposition (19, 24, 25). Reduced *PRRX1* expression in human cardiac tissues has been associated with AF (19). We previously deleted the mouse orthologous variant region near *PRRX1* (*Prrx1(enh)*) to investigate its role in gene regulation and rhythm control (25). *Prrx1(enh)^-/-^* mice express less *Prrx1* specifically in cardiomyocytes compared to controls, and show atrial conduction slowing, lower AP upstroke velocity (indicative for lower Na^+^ current densities) (58), as well as increased systolic and diastolic [ca^2+^]_i_ concentration that culminate in increased susceptibility to atrial arrhythmia induction (25). To explore whether and how these two AF-risk genes may interact, we intercrossed *RE(int)^-/^*^-^ with *Prrx1(enh)^-/-^* mice (decreased *Prrx1* in cardiomyocytes), and investigated cardiac transcriptomes and phenotypes across genotypes.”

How do the authors explain that lack of change noted for Pitx2?

We did not observe a change in *Pitx2* expression in the left atria of our mutant mice. We surmise that this is because a 30-35% increase of *Tbx5* expression is not sufficient to induce a proportional increase in *Pitx2*. We recently characterized a *Tbx5* gain-of-function mouse model (van Ouwerkerk et al., 2022) in which *Pitx2* was increased in the right atrium, while the left atrium remained unaffected. We therefore examined whether *Pitx2* expression was changed in the right atria of RE(int)^-/-^ by RT-qPCR and found that it is indeed not affected, thus suggesting that the modest increase in *Tbx5* level does not influence *Pitx2* expression.

We included this result in lines 215-216, new Figure 5—figure supplement 3, and discussion in lines 348-356.

Reviewer #2 (Recommendations for the authors):The rationale for focusing on RE(down) is not clearly appreciated based on Figure 1. This is in contrast to the apparent evidence in support of RE(int). Promoter Hi-C data shows DNA interactions in human that appears to span beyond the presumptive syntenic region of RE(down) in mouse, as well as a lack of chromatin accessibility, histone marks or EMERGE prediction for RE(down). Perhaps highlighting relevant features for RE(down) more explicitly would address this minor concern.

We would like to thank the reviewer for this suggestion. While the absence of interactions as assayed in Promoter Capture Hi-C does not necessarily indicate a lack of interaction, the interactions beyond the RE(down) region indeed indicate possible distal downstream regulatory elements. Consistently, Smemo et al. (Smemo et al., 2012) identified multiple enhancer candidates downstream of the REdown region, one of which was found to harbor a SNP associated with CHD (non-HOS). We chose to include the RE(down) region in our study because it harbors AF-associated variants, and H3K4me1 modifications in cardiomyocytes. We have clarified this in lines 125-126 of the revised manuscript.

For supplemental figure 1, please define LD as "linkage disequilibrium" prior to abbreviation, as appropriate, and list the abbreviation at the beginning of the manuscript. As well, a scale bar is not apparent.

We thank the reviewer for pointing out these oversights. We have changed the text in the supplement as indicated.

Does the increase in Tbx5 gene expression result in an increase in protein expression?

We attempted to quantify protein expression in adult atria of RE(int)^-/-^ and WT mice by Western Blot. However, we were unable to detect Tbx5 protein. We performed immunofluorescence staining on adult atria to show that indeed Tbx5 protein is detected in the atria of mutant and control mice, even though a 30% increase is not quantifiable using this method. However, the analysis shows that Tbx5 remains selectively expressed in PCM-1^+^ cardiomyocyte nuclei in RE(int)^-/-^ atria (see new Figure 2—figure supplement 2).

It would be interesting to compare these mouse models to the recently published TBX5 gain of function model.

We thank the reviewer for this valuable suggestion. We now include an additional table and a scatter plot showing a comparison between adult left atria from RE(int)^-/-^ and Tbx5^G125R/+^ as Supplementary file 4 and in Figure 5—figure supplement 2, respectively. Surprisingly, we found very few differentially expressed genes (DEGs) common to both models, while their electrophysiological phenotypes were more alike- inverted P-waves, slower and more variable heart rates, increased atrial arrhythmia induction, and prolonged APD. Conspicuously, an inducible loss-of-function *Tbx5* mouse model (Nadadur et al., 2016) also displays irregular RR, prolonged APD and arrhythmia. We have also compared transcriptomes of left atria of our RE(int) model to those of this induced adult *Tbx5* deletion model (Supplementary file 5 and in Figure 5—figure supplement 2) and found opposite response of many of the same DEGs, as expected. Together, this suggests that perturbation of a Tbx5–dependent network independently of direction of change of many of the network genes results in overlapping phenotypic changes culminating in atrial dysfunction.

We have included this comparison in lines 204-208.

For Figure 3, please define RR as "R-R interval" in legend and SDNN as "standard deviation of normal to normal R-R intervals", as appropriate, and add to abbreviation list.

We apologize for the oversight. The definitions have been added to the revised manuscript as indicated.

In Figure 5, where is Prrx1 highlighted in the transcriptomic analysis?

*Prrx1* was, in fact, not found to be differentially expressed in left atrial RNAseq data. We found that the induction of *Prrx1* occurs only in atrial cardiomyocytes of RE(int)^-/-^ mice, consistent with cardiomyocyte-specific expression-, and thereby induction in RE(int)^-/-^, of *Tbx5* (see also immunofluorescence analysis mentioned above). We surmise that this change was not detectable in bulk RNAseq because *Prrx1* is mostly expressed in non-cardiomyocytes, which is likely not affected in our mutants.

We understand that the rationale for studying the interaction between *Tbx5* and *Prrx1* is muddled without this observation. Therefore, we have changed the text in lines 229-238, as specified in reviewer 1 point 5.

For Figure 6C, how does Prrx1(enh)-/- compare in the PCA analysis? Please display it.

Unfortunately, we are unable to directly compare the Prrx1(enh)^-/-^ RNAseq dataset in the PCA because it was obtained, along with its own WT control group, in a different session (months in between experiments; all experiments performed with alleles on same FVB background though). It appeared that this difference in session greatly influences the PCs. However, we compared the L2FC (and p(adj)) values in each transcriptomic comparison (WT vs RE(int)^-/-^, WT vs Prrx1(enh)^-/-^, WT vs RE(int)^-/-^;Prrx1(enh)^-/-^ (Double mutant)), as these parameters are relatively unaffected by inter-session variation. We observed that (1) the number of genes differentially expressed between mutant and WT decreased going from RE(int)^-/-^, Double mutant, Prrx1(enh)^-/-^, (2) the direction of change of DEGs was similar between RE(int)^-/-^ and Double mutant, but different from Prrx1(enh)^-/-^, (3) the distribution of the L2FCs was slightly smaller in Double mutant than RE(int)^-/-^. This is shown in the revised Supplemental figure 10. Together, these analyses suggest that RE(int)^-/-^ has a much larger effect on the atrial transcriptome than Prrx1(enh)^-/-^, and that the effect of RE(int)^-/-^ is slightly normalized by the inclusion of Prrx1(enh)^-/-^ in RE(int)^-/-^.

Reviewer #3 (Recommendations for the authors):1. TBX5 mutations in humans and animal models are known to generate congenital heart defects and upper limb/radial malformations. Perhaps of relevance to this manuscript, human patients with TBX5 duplications also show many of these same phenotypes (Heinritz et al. 2005 Heart; Patel et al. 2012 Eur J Hum Genet; Kimura et al. 2015 Pedatr Cardiol; Cenni et al. 2021 Eur J Hum Genet). Do RE(int) and RE(down) homozygous mutations result in congenital heart or limb malformations? Structural heart defects can certainly impact cardiac electrophysiology and reporting structural defects, or lack of them can help put the phenotype into context.

We thank the reviewer for this suggestion. We have included the notion of CHD in patients with duplications and these references in the Introduction section of the revised manuscript in lines 85-86. We did not observe any structural defects of limbs or heart. *Tbx5* expression in RE(down)^-/-^ mice is selectively induced in atrial and lung tissue after birth, and therefore limb defects are not expected. Similarly, atrial induction in RE(int)^-/-^ mice is observed after birth. However, we noted a mild induction of *Tbx5* expression in prenatal ventricles of RE(int)^-/-^ mice (not in RE(down)^-/-^ mice). While this does not lead to structural defects, the ventricles may be slightly larger. We are currently investigating this possible phenotype.

2. The RE(int) deletion appears to be more severe than the RE(down) deletion. Does the RE(int) deletion interfere with the splicing of the last two exons? Depending on library construction, perhaps the RNA-seq can address this.

This is a very insightful suggestion. Inspection of the RNAseq data sets revealed that the exon read coverage of exons relative to each other, including the ones flanking the intra-intronic deletion, is highly similar between WT and RE(int)^-/-^ (see new Figure 2—figure supplement 3). Moreover, when zooming in on the intronic region, we clearly see absence of reads mapped to the deleted region, validating the absence of this genomic sequence. Furthermore, when we performed RT-qPCR of WT and RE(int)^-/-^ atrial samples using intron/exon-spanning primers in exons 6-8, 7-9 and 8-9, respectively, we obtained the same result for both groups, and the obtained amplicons did not include fragments in which exon 8 was skipped (new Figure 2—figure supplement 3).

3. In Figure 2E, the authors report that rs7312625 A>G appears to be the functional SNP as it alone results in increased Tbx5 expression in vitro. As this is a predicted regulatory element, what transcription factor binding motif(s) is this SNP predicted to interfere or generate? Does this motif change make sense with gained Tbx5 expression in cardiomyocytes/HL1 cells?

We analyzed whether the rs7312625 substitution interfered/changed predicted binding motifs and found that the risk allele disrupts motifs for SOL1/2 (TCX/TCX2/CHC), which are homologues of the animal CHC protein LIN54 that binds DNA in a sequence-specific manner and is a component of DREAM complexes regulating cell cycle-dependent transcription. Furthermore, we observed gain of a *sine oculis* (SIX) homeodomain transcription factor binding motif with the risk allele (new Supplementary file 1). SIX transcription factors can act as transcriptional activators or repressors depending on interactions with other highly conserved regulators including Paired-box (Pax), Eyes absent (Eya), Dachshund (Dach), and Groucho (Grg) proteins (Meurer et al., 2021). Both Six4 and Six5 are expressed in atrial tissue of mice (see Supplementary table 1) and human (see (Meurer et al., 2021)). We speculate the SNP could create a SIX binding site, causing loss of repressive activity of the RE. We have added this information to the revised manuscript in lines 152-159 of the Results section and 295-298 in the discussion.

4. In Figure 3C, the authors report RE(int) mutants have an increased PR interval and in Figure 4C, the authors report RE(int) have an increased APD20, 50, and 90. Single-copy loss of Tbx5 in mice has been reported to cause these same phenotypes (e.g. Bruneau et al. 2001 Cell; Nadadur et al. 2016 Sci Trans Med). Would you not expect the opposite phenotype, i.e., decreased PR interval and APD20/50/90? What is the explanation for these paradoxical results?

This observation is puzzling to us as well, and may reflect the complexity of responses to either decreased or increased levels of *Tbx5*. When comparing the transcriptional response of increased *Tbx5* in left atria (RE(int)^-/-^) to that of Tbx5^G125R/+^ (gain of function missense mutation; (van Ouwerkerk et al., 2022) (new Supplementary file 4, Figure 5—figure supplement 2)) we observed that it was unexpectedly divergent. When comparing the published left atrial transcriptomes of Tbx5 iKO ((Nadadur et al., 2016); *Tbx5* deletion induced in adult mice) with that of RE(int)^-/-^, we found, as expected, that most significantly differentially expressed genes (DEGs) common to both datasets were changed in opposite direction (new Supplementary file 5, Figure 5—figure supplement 2). Remarkably, all three *Tbx5* mouse models display APD prolongation. This suggests that a (probably) large number of transcriptional changes occur in RE(int)^-/-^ that disturb the balance between AP duration-increasing and -decreasing gene expression, leading to phenotypic outcome similar to that of both *Tbx5* loss of function and Tbx5 G125R missense variant. This further indicates that the response to Tbx5 dose is complex. Moreover, as indicated by the reviewer, loss- and gain of TBX5 gene copy results in CHD phenotypes, consistent with the notion that both reduced and increased Tbx5 levels can result in similar phenotypic outcomes.

5. Several transcriptional profiling datasets have been generated for various Tbx5 mutations in mouse and human including a few by the authors (by cursory search of GEO for "TBX5"). How do the gene expression changes seen in the current datasets compare with previously generated datasets? Are the changes largely consistent with expectations of increased Tbx5 levels in RE(int) and RE(down)? It appears most public datasets are in Tbx5/TBX5 loss-of-function contexts, so we are expecting the opposite results. Does this data support that hypothesis or is there something else going on?

We have included a new supplemental figure and accompanying tables in which we compare transcriptional profiles of left atria of RE(int)^-/-^ mice to those of a Tbx5 gain-of-function variant mouse model (van Ouwerkerk et al., 2022) and an adult-specific knockout mouse model (Nadadur et al., 2016). Please see reviewer 2 point 4, and the response this reviewer above.

6. Figures 2A and 2C report that RE(down) demonstrated increased expression of Tbx5 in the adult left atria and P21 lungs, specifically, but there is little to no mention of this in the text or discussion. Given the importance of the pulmonary veins in the development of atrial fibrillation and prior publication history from the authors and colleagues on the topic of left atrial Tbx5, can the authors expand their discussion on the RE(down) related SNPs? The SNPs associated with RE(down) are independently segregating and the differences observed in this manuscript may hint at an independent, tissue-dependent function for that regulatory element.

This is an excellent suggestion. We have addressed this point in the Discussion lines 302-313 as follows:

“…the AF-associated variants in *RE(down)* may increase *Tbx5* expression in both the left atrium and pulmonary vein myocardium. The pulmonary veins have been strongly implicated in AF as they are the most common source of triggered activity (64, 65). *TBX5* levels in pulmonary vein myocardium may influence gene regulation independently from the atrial *TBX5* levels (66). The variants in *RE(down)* segregate independently from those in *RE(int)*, indicating these variant RE regions act through distinct tissue-specific transcriptional mechanisms that are influenced by AF-predisposing common variants. A further implication is they may cumulatively increase AF predisposition in a manner dependent on risk variant dose, in which homozygous carriers of risk haplotypes in *RE(int)* and *RE(down)* have the largest relative predisposition. These relations could be addressed in future genotype-phenotype analyses of human atrial and pulmonary vein samples.”

References

Dixon, J.R., Selvaraj, S., Yue, F., Kim, A., Li, Y., Shen, Y., Hu, M., Liu, J.S., Ren, B., 2012. Topological domains in mammalian genomes identified by analysis of chromatin interactions. Nature 485, 376–80. https://doi.org/10.1038/nature11082

Meurer, L., Ferdman, L., Belcher, B., Camarata, T., 2021. The SIX Family of Transcription Factors: Common Themes Integrating Developmental and Cancer Biology. Front. Cell Dev. Biol. 9.

Nadadur, R.D., Broman, M.T., Boukens, B., Mazurek, S.R., Yang, X., Van Den Boogaard, M., Bekeny, J., Gadek, M., Ward, T., Zhang, M., Qiao, Y., Martin, J.F., Seidman, C.E., Seidman, J., Christoffels, V., Efimov, I.R., McNally, E.M., Weber, C.R., Moskowitz, I.P., 2016. Pitx2 modulates a Tbx5-dependent gene regulatory network to maintain atrial rhythm. Sci. Transl. Med. 8. https://doi.org/10.1126/scitranslmed.aaf4891

Nora, E.P., Lajoie, B.R., Schulz, E.G., Giorgetti, L., Okamoto, I., Servant, N., Piolot, T., Van Berkum, N.L., Meisig, J., Sedat, J., Gribnau, J., Barillot, E., Blüthgen, N., Dekker, J., Heard, E., 2012. Spatial partitioning of the regulatory landscape of the X-inactivation centre. Nature 485, 381–385. https://doi.org/10.1038/nature11049

Sexton, T., Yaffe, E., Kenigsberg, E., Bantignies, F., Leblanc, B., Hoichman, M., Parrinello, H., Tanay, A., Cavalli, G., 2012. Three-dimensional folding and functional organization principles of the *Drosophila* genome. Cell 148, 458–472. https://doi.org/10.1016/j.cell.2012.01.010

Smemo, S., Campos, L.C., Moskowitz, I.P., Krieger, J.E., Pereira, A.C., Nobrega, M.A., 2012. Regulatory variation in a TBX5 enhancer leads to isolated congenital heart disease. Hum. Mol. Genet. 21, 3255–63. https://doi.org/10.1093/hmg/dds165

van Ouwerkerk, A.F., Bosada, F.M., van Duijvenboden, K., Houweling, A.C., Scholman, K.T., Wakker, V., Allaart, C.P., Uhm, J.-S., Mathijssen, I.B., Baartscheer, T., Postma, A.V., Barnett, P., Verkerk, A.O., Boukens, B.J., Christoffels, V.M., 2022. Patient-Specific TBX5-G125R Variant Induces Profound Transcriptional Deregulation and Atrial Dysfunction. Circulation 145, 606–619. https://doi.org/10.1161/CIRCULATIONAHA.121.054347

Weerd, J.H. van, Badi, I., Boogaard, M. van den, Stefanovic, S., Werken, H.J.G. van de, Gomez-Velazquez, M., Badia-Careaga, C., Manzanares, M., Laat, W. de, Barnett, P., Christoffels, V.M., 2014. A Large Permissive Regulatory Domain Exclusively Controls Tbx3 Expression in the Cardiac Conduction System. Circ. Res. 115, 432–441. https://doi.org/10.1161/CIRCRESAHA.115.303591